# DisPPO: Quantile-Based Distributional Reinforcement Learning for Large Language Models

**Zhijian Zhou** [* 1 2 3]  **Long Li** [* 4]  **Xuan Zhang** [* 1 2 3]  **Zongkai Liu** [2]  **Yanting Miao** [5]  **Yuchen Liu** [1]  **Deshu Chen** [1]
**Ke Li** [3]  **Xing Sun** [3]  **Ruoxi Jiang** [† 1 6]  **Xiaoyu Tan** [† 3]  **Chao Qu** [†‡ 1 6]  **Yuan Qi** [† 1 6]

## Abstract

Reinforcement Learning (RL) has become a cornerstone for enhancing the reasoning capabilities of Large Language Models (LLMs). However, standard actor-critic methods, such as PPO, rely on scalar value functions that estimate only the expectation of cumulative returns. This reduction inherently discards higher-order statistical information (e.g., variance and multimodality), leading to inaccurate value estimation and suboptimal credit assignment in complex tasks. While Distributional RL offers a solution by modeling the full return distribution, its application to LLMs remains challenging due to the computational intractability of value-based operations over large vocabularies and the optimization instability of off-policy replay mechanisms. In this paper, we propose **DisPPO**, a novel on-policy framework that seamlessly integrates non-parametric quantile regression into PPO. Theoretically, we prove that our distributional update operator—composed of the $\lambda$-return Bellman operator and quantile projection—is a contraction mapping in the Wasserstein metric, guaranteeing convergence to a unique fixed point. Empirically, we evaluate DisPPO using Llama and Qwen models across mathematical reasoning and Text-to-SQL benchmarks, where it consistently outperforms standard PPO and group-based baselines.

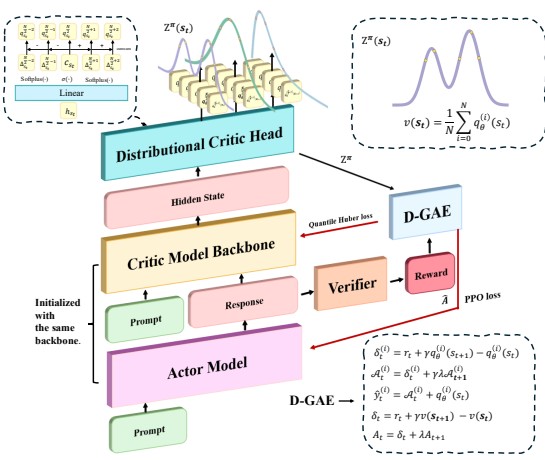

*Figure 1.* Overview of DisPPO . The actor maps a prompt to a response. The critic takes the prompt–response pair and, using a critic backbone that is architecturally identical to the actor and initialized with the same parameters, produces a hidden representation $h(s_t)$ of the current state $s_t$. These hidden states are fed into a distributional critic head, which outputs a return distribution $Z^\pi(s_t)$ represented by an ordered set of quantiles $\{q_\theta^{(n)}(s_t)\}$, rather than a scalar value $V(s_t)$ as in standard PPO. The quantiles are trained via quantile Huber regression and are used to construct value targets and advantages through Distributional Generalized Advantage Estimation (D-GAE). D-GAE produces (i) multi-step regression targets $\hat{y}_t$ for distributional critic learning and (ii) scalar advantages $\hat{A}_t$ for policy optimization. Black arrows denote inference, while red arrows indicate training. Dashed boxes illustrate the center–delta quantile head (top left) and the D-GAE formulation (bottom right).

[*]Equal contribution  [1]Fudan University, Shanghai, China [2]Shanghai Innovation Institute, Shanghai, China [3]Tencent Youtu Lab, Shanghai, China [4]Griffith University, Queensland, Australia [5]University of Waterloo, Ontario, Canada [6]Shanghai Academy of Artificial Intelligence for Science, Shanghai, China. Correspondence to: Ruoxi Jiang <roxie_jiang@fudan.edu.cn>, Xiaoyu Tan <arthurtan@tencent.com>, Chao Qu <quchao@fudan.edu.cn>, Yuan Qi <qiyuan@fudan.edu.cn>.

*Proceedings of the $43^{rd}$ International Conference on Machine Learning*, Seoul, South Korea. PMLR 306, 2026. Copyright 2026 by the author(s).

[†]Corresponding author.

[‡]Project leader.

Contact: `zjzhou24@m.fudan.edu.cn`.

## 1. Introduction

Large language models (LLMs) have demonstrated an unprecedented capacity for complex reasoning tasks, largely enabled by the Chain-of-Thought (CoT) paradigm (Wei et al., 2022; OpenAI, 2024). To unlock the full potential of these models, Reinforcement Learning (RL) has proven pivotal, leveraging verifiable outcome signals to guide structured exploration (Shao et al., 2024). Current RL strategies for LLMs encompass a diverse spectrum, ranging from group-based outcome optimization methods like GRPO and DAPO (Shao et al., 2024; Yu et al., 2025) to step-wise actor-critic frameworks such as PPO (Schulman et al., 2017;

Ouyang et al., 2022). Among these, Actor-Critic frameworks remain a research and deployment cornerstone due to their capacity for dense credit assignment and granular control over the generation process. However, the efficacy of these methods hinges critically on the critic's ability to *accurately* estimate the value of long, complex reasoning trajectories (Yue et al., 2025). Since the value function serves as a baseline to reduce the variance of policy gradient updates, any inaccuracy in the critic manifests as noisy advantage estimates, leading to unstable training and suboptimal convergence (Sutton et al., 1998).

In standard actor–critic RL, the critic predicts a single scalar value $V^\pi(s)$, trained via mean squared error (MSE) to approximate the expected return from a state.

While effective in many settings, this approach with expected return compresses all uncertainty about the underlying stochasticity of the future outcomes into a single number.

As a result, important statistical information about the distribution of the returns, such as variability, multimodality, is discarded (Bellemare et al., 2017).

This loss of information is particularly acute in reasoning tasks, where rewards are often sparse or binary.

Distributional Reinforcement Learning (DistRL) addresses this limitation by modeling the entire probability distribution of returns, capturing richer learning signals that have led to superior representation learning in both game-playing and continuous control domains (Dabney et al., 2018b; Hessel et al., 2018; Barth-Maron et al., 2018).

Despite this promise, directly applying existing DistRL methods to LLM training faces several practical and conceptual challenges. First, classic value-based distributional methods (e.g., QR-DQN (Dabney et al., 2018b)) rely on an $\arg\max_a Q(s, a)$ operation, which are computationally prohibitive given the vast vocabulary size of LLMs. Second, off-policy actor-critic algorithms like D4PG (Barth-Maron et al., 2018) and DSAC (Ma et al., 2025), while effective in robotics, typically rely on extensive experience replay mechanisms to stabilize training. Maintaining such massive buffers for LLMs is memory-prohibitive, and the use of stale off-policy data often introduces optimization instability compared to the on-policy updates. Third, prior attempts to integrate distributional critics into policy gradient methods often diverge in their core objectives or impose restrictive assumptions. For example, works like WCPG and Risk-averse SDPG (Tang et al., 2020; Singh et al., 2020) utilize return distributions to construct risk-sensitive objectives (e.g., optimizing CVaR (Rockafellar et al., 2000)). While valuable for safety, this alters the standard reinforcement learning goal of maximizing expected correctness required for reasoning tasks. Conversely, methods that retain the standard objective, such as GMAC (Nam et al., 2021), impose parametric

assumptions (e.g., Gaussian Mixture Models) that fail to capture the sparse, multi-modal return landscapes of reasoning trajectories. Moreover, beyond these representational limitations, such parametric extensions often lack rigorous theoretical analysis regarding the contraction properties of their underlying distributional operators.

In this work, we propose **DisPPO**, a novel algorithm that seamlessly integrates non-parametric quantile regression with PPO. Unlike prior methods, DisPPO retains the memory efficiency of on-policy updates while leveraging a distributional critic to learn robust state representations. Theoretically, we prove that our composite update operator acts as a contraction mapping in the Wasserstein metric, guaranteeing a unique fixed point for the value distribution. Crucially, we investigate the fundamental question of value estimation accuracy in LLMs. Our empirical analysis in section 6.3 shows that estimating the full return distribution and subsequently deriving the mean yields a *more accurate value estimate* against ground-truth Monte Carlo returns compared to *directly fitting the mean*. This suggests that the distributional objective provides a superior supervisory signal for value function approximation.

Our main contributions are summarized as follows:

- **Distributional PPO for LLMs (DisPPO).** We propose DisPPO, a novel on-policy reinforcement learning framework that seamlessly integrates non-parametric quantile regression into the PPO critic. By learning the full return distribution, DisPPO produces more accurate and stable value estimates than standard scalar critics, while introducing only negligible training-time overhead compared to vanilla PPO (see Section 6.1).

- **Theoretical Convergence Guarantee.** We provide a rigorous theoretical analysis of the underlying distributional operator. We prove that the composition of the distributional $\lambda$-Bellman operator and the quantile projection constitutes a contraction mapping under the Wasserstein metric.

- **Superior Performance and Robustness.** We conduct extensive experiments with Qwen and Llama series models across three challenging domains: mathematical reasoning, SQL code generation, and multi-turn tool-augmented reasoning. DisPPO consistently outperforms standard PPO, recent group-based methods (GRPO, DAPO), and their variants in both Avg@k and Pass@k metrics.

Our code is publicly available at https://github.com/ZhijianZhou/Disppo.

## 2. Preliminaries

In this section, we establish the notation for Markov Decision Processes and Distributional Reinforcement Learning. We further review the Wasserstein metric and Quantile Regression, which form the theoretical and practical foundations of our method.

### 2.1. Distributional Reinforcement Learning

We consider a Markov Decision Process (MDP) defined by the tuple $(\mathcal{S}, \mathcal{A}, P, R, \gamma)$, where $\mathcal{S}$ and $\mathcal{A}$ denote the state and action spaces, $P(\cdot|s, a)$ is the transition kernel, $R(s, a)$ is the reward function, and $\gamma \in [0, 1)$ is the discount factor.

Let $Z^\pi(s, a) = \sum_{t=0}^{\infty} \gamma^t R(s_t, a_t)$ denote the random variable representing the cumulative discounted return obtained by following policy $\pi$ starting from state $s$ and action $a$. Standard Reinforcement Learning (RL) seeks to estimate the expected value of this return, i.e., the scalar value function $Q^\pi(s, a) = \mathbb{E}[Z^\pi(s, a)]$.

Distributional RL, conversely, explicitly models the full distribution of the random return $Z^\pi(s, a)$ (Bellemare et al., 2017). The distribution of $Z^\pi$ is characterized by the *distributional Bellman equation*:

$$Z^\pi(s, a) \overset{D}{=} R(s, a) + \gamma Z^\pi(s', a'), \qquad (1)$$
$$s' \sim P(\cdot|s, a), \ a' \sim \pi(\cdot|s').$$

where $\overset{D}{=}$ denotes equality in distribution. This gives rise to the distributional Bellman operator $\mathcal{T}^\pi$, acting on the space of return distributions $\mathcal{Z}$, such that the distribution of $Z^\pi$ is the fixed point $\mathcal{T}^\pi Z^\pi = Z^\pi$.

### 2.2. The Wasserstein Metric

To analyze the convergence of distributional operators (such as the $\mathcal{T}^{\lambda,\pi}$ introduced in Section 3), we require a metric on the space of probability distributions. The $p$-Wasserstein metric $W_p$ (for $p \in [1, \infty]$) between two distributions $U$ and $Y$ is defined as the $L^p$ distance between their inverse cumulative distribution functions (Müller, 1997) (inverse CDFs):

$$W_p(U, Y) = \left( \int_0^1 \left| F_Y^{-1}(\omega) - F_U^{-1}(\omega) \right|^p d\omega \right)^{1/p}, \quad (2)$$

where $F_Y^{-1}(\omega) := \inf\{y \in \mathbb{R} : \omega \leq F_Y(y)\}$ is the inverse CDF (or quantile function) of $Y$.

To establish the contraction properties of Bellman operators over the entire state-action space, we utilize the maximal Wasserstein metric $\bar{d}_p$ (Bellemare et al., 2017), defined as the supremum of the Wasserstein distance over all state-action pairs[1]:

$$\bar{d}_p(Z_1, Z_2) := \sup_{(s,a) \in \mathcal{S} \times \mathcal{A}} W_p(Z_1(s, a), Z_2(s, a)). \quad (3)$$

As shown by Bellemare et al. (2017), the distributional Bellman operator is a $\gamma$-contraction under $\bar{d}_p$, guaranteeing a unique fixed point $Z^\pi$. Consequently, the fundamental objective of distributional RL is to find a parameterized distribution $Z_\theta$ that minimizes the Wasserstein error against the Bellman target: $\min_\theta W_p(Z_\theta, \mathcal{T}^\pi Z_\theta)$. However, since the space of all probability measures is infinite-dimensional, directly optimizing this objective is computationally intractable. This necessitates projecting the target distribution onto a finite-dimensional manifold, as we detail next.

### 2.3. Quantile Regression and QR-DQN

To make distribution learning computationally tractable, QR-DQN (Dabney et al., 2018b) approximates the return distribution using a parametric family of distributions $\mathcal{Z}_N \subset \mathcal{Z}$. Specifically, we consider the space of uniform mixtures of $N$ Diracs supported on learnable locations $\{q_i\}_{i=1}^N$:

$$\mathcal{Z}_N := \left\{ Z_\theta = \frac{1}{N} \sum_{i=1}^N \delta_{q_i} \mid q_1 \leq q_2 \leq \cdots \leq q_N \right\}. \quad (4)$$

To map an arbitrary target distribution $Y$ back onto this manifold $\mathcal{Z}_N$, the 1-Wasserstein Projection operator $\Pi_{W_1}$ is defined as :

$$\Pi_{W_1} Y := \underset{Z_\theta \in \mathcal{Z}_N}{\arg\min}\, W_1(Y, Z_\theta). \quad (5)$$

Dabney et al. (2018b) showed that finding the projected distribution $\Pi_{W_1} Y$ is asymptotically equivalent to minimizing the *quantile regression loss* on samples from $Y$.

In QR-DQN, the target distribution is constructed by applying the distributional Bellman operator to the quantiles of the next state. For a transition $(s, a, r, s')$, the set of target support points $\{\hat{y}^{(j)}\}_{j=1}^N$ is defined as:

$$\hat{y}^{(j)} = r + \gamma q_{\theta'}^{(j)}(s', a'), \quad j = 1, \ldots, N, \quad (6)$$

where $q_{\theta'}$ denotes the target network. To facilitate gradient-based optimization, the non-smooth quantile loss is replaced by the Quantile Huber loss (Huber, 1992). The total objective minimizes the distance between the predicted quantiles and these targets:

$$\mathcal{L}_{\text{QR-DQN}} = \frac{1}{N^2} \sum_{i=1}^N \sum_{j=1}^N \rho_{\tau_i}^\kappa \left( \hat{y}^{(j)} - q_\theta^{(i)}(s, a) \right), \quad (7)$$

---

[1] While we follow the standard notation $Z(s, a)$ from value-based literature (Bellemare et al., 2017) in this section, our method (DisPPO) estimates the state-value distribution $Z(s)$. The metric and theoretical properties discussed here apply analogously to $Z(s)$ by taking the supremum over $\mathcal{S}$.

where $\tau_i = \frac{2i-1}{2N}$ are the cumulative probabilities, and $\rho_\tau^\kappa$ is the asymmetric Huber loss:

$$\rho_\tau^\kappa(u) = |\tau - \mathbb{I}(u < 0)| \cdot \begin{cases} \frac{1}{2}u^2, & |u| \leq \kappa, \\ \kappa(|u| - \frac{1}{2}\kappa), & \text{otherwise.} \end{cases} \quad (8)$$

# 3. Method

In this section, we present DisPPO, a framework that integrates distributional reinforcement learning into the on-policy PPO paradigm. As illustrated in Figure 1, our framework replaces the standard scalar critic with a quantile-based distributional critic while maintaining the efficiency of the PPO actor. This architecture allows the model to capture the complex return distribution of reasoning tasks without altering the fundamental trust-region optimization of the policy. We detail the construction of our distributional critic, the practical Distributional Generalized Advantage Estimation (D-GAE), and the corresponding actor update mechanism.

## 3.1. From Scalar to Distributional Targets

**The Scalar Perspective.** In standard PPO, the critic estimates the scalar value function $V^\pi(s) = \mathbb{E}[Z^\pi(s)]$. A central challenge in value estimation is balancing the trade-off between bias (from single-step bootstrapping) and variance (from full Monte Carlo returns). To address this, PPO utilizes the Generalized Advantage Estimation (GAE) (Schulman et al., 2016), which effectively constructs a target based on the $\lambda$-return. Conceptually, this target is an geometrically weighted average of all $n$-step returns, allowing the algorithm to smoothly interpolate between TD(0) and Monte Carlo targets. Formally, this corresponds to applying the *scalar $\lambda$-operator* $\mathcal{T}_V^{\pi,\lambda}$, defined as the geometric mixture of $n$-step Bellman operators:

$$\mathcal{T}_V^{\pi,\lambda} V(s) := (1 - \lambda) \sum_{n=0}^{\infty} \lambda^n (\mathcal{T}^\pi)^{n+1} V(s). \quad (9)$$

**The Distributional Perspective.** Our method shifts the critic's objective to model the full random variable $Z^\pi(s)$. We extend the intuition of the scalar $\lambda$-return to the distributional setting by constructing a target that is the geometric mixture of distributional $n$-step updates. We formally define the **distributional $\lambda$-Bellman operator** $\mathcal{T}^{\pi,\lambda}$:

$$\mathcal{T}^{\pi,\lambda} Z(s) := (1 - \lambda) \sum_{n=0}^{\infty} \lambda^n (\mathcal{T}^\pi)^{n+1} Z(s). \quad (10)$$

This operator acts as the theoretical target for our method. As we rigorously prove in Section 4, this operator is a contraction mapping, ensuring a unique fixed point for the return distribution.

## 3.2. Critic Update via Distributional GAE

Since the infinite sum in Eq. (10) is intractable, we propose **Distributional GAE (D-GAE)** to approximate this target using sampled trajectories. This process involves three steps: parameterization, target estimation, and projection.

**Parameterization.** Following the Quantile Regression framework introduced in Section 2, we approximate the value distribution $Z^\pi(s)$ using a discrete set of $N$ quantiles $\{q_\theta^{(i)}(s)\}_{i=1}^N$. These quantiles correspond to fixed cumulative probabilities $\tau_i = \frac{2i-1}{2N}$, effectively acting as a learnable non-parametric representation of the inverse CDF. Crucially, to constitute a valid inverse CDF, these estimates must satisfy the monotonicity property $q_\theta^{(1)}(s) < q_\theta^{(2)}(s) < \cdots < q_\theta^{(N)}(s)$. Violating this "non-crossing" constraint can lead to optimization instability (Zhou et al., 2020). Therefore, architecturally, we enforce strict ordering by parameterizing the critic head to predict a base quantile and a sequence of non-negative deltas, which are cumulatively aggregated to form the ordered quantiles. Detailed architecture specifications are provided in Appendix D.1.

**Target Estimation via D-GAE.** To estimate the target distribution defined by the $\lambda$-operator $\mathcal{T}^{\pi,\lambda}$, we extend the Generalized Advantage Estimation (GAE) to the distributional setting. Standard GAE computes a scalar shift to the value estimate. In D-GAE, we apply this shift *element-wise* to each quantile location. For a sampled trajectory $\tau = (s_0, a_0, r_0, s_1, a_1, \ldots, s_T)$, we first compute the temporal difference error for each quantile level $i$:

$$\delta_t^{(i)} = r_t + \gamma q_\theta^{(i)}(s_{t+1}) - q_\theta^{(i)}(s_t). \quad (11)$$

By accumulating these errors geometrically (analogous to the scalar $\lambda$-return), we derive the *distributional advantage* for each quantile:

$$\mathcal{A}_t^{(i)} = \delta_t^{(i)} + (\gamma\lambda)\mathcal{A}_{t+1}^{(i)}. \quad (12)$$

The regression target for the $i$-th quantile, $\hat{y}_t^{(i)}$, is then constructed by shifting the current estimate by this advantage:

$$\hat{y}_t^{(i)} = q_\theta^{(i)}(s_t) + \mathcal{A}_t^{(i)}. \quad (13)$$

Intuitively, $\hat{y}_t^{(i)}$ represents a sample-based approximation of the distribution after applying the contraction operator $\mathcal{T}^{\pi,\lambda}$.

**Projection via Quantile Regression.** Finally, we must project this updated target distribution back onto our parameterized quantile manifold. As defined in Section 2, we project the updated target distribution using the 1-Wasserstein projection. Recall that minimizing the Quantile Regression loss is asymptotically equivalent to minimizing

this Wasserstein distance. Therefore, to update the critic parameters $\theta$, we minimize the Quantile Huber Loss averaged over trajectories $\xi$ of length $T$ sampled from the current policy $\pi$. Specifically, we compute the discrepancy between our predicted quantiles $q_\theta^{(i)}(s_t)$ and the constructed targets $\hat{y}_t^{(i)}$:

$$\mathcal{L}_{\text{critic}}(\theta) = \mathbb{E}_{\xi \sim \pi} \left[ \frac{1}{T} \sum_{t=0}^{T-1} \frac{1}{N^2} \sum_{i=1}^{N} \sum_{j=1}^{N} \right. \qquad (14)$$
$$\left. \rho_{\tau_i}^\kappa \left( \hat{y}_t^{(j)} - q_\theta^{(i)}(s_t) \right) \right],$$

where $\rho_\tau^\kappa$ is the asymmetric Huber loss defined in Eq. (8). This ensures that our critic update performs a valid Wasserstein projection, preserving the contraction properties of the Bellman operator.

### 3.3. Actor Update

Finally, we perform the actor update using the value estimates derived from our distributional critic. To compute the Generalized Advantage Estimation (GAE) for the PPO policy gradient, we first extract a scalar state-value estimate $v(s_t)$ by averaging our predicted quantiles:

$$v(s_t) = \frac{1}{N} \sum_{i=1}^{N} q_\theta^{(i)}(s_t). \qquad (15)$$

**Risk-Neutral Policy Optimization.** It is important to note that while our critic models the full return distribution, the actor continues to maximize the expected return via the scalar mean $v(s_t)$. This design choice aligns with established distributional RL methodologies, such as QR-DQN (Dabney et al., 2018b) and D4PG (Barth-Maron et al., 2018), which demonstrate that the primary performance gains of DistRL stem from the richer training signal and improved representation learning provided by the distributional auxiliary task, rather than from altering the policy's decision criterion (e.g., to be risk-averse). Consequently, we maintain the standard objective to optimize for reasoning accuracy without introducing bias from risk-sensitive metrics.

Using this distribution-derived scalar value, we compute the standard scalar advantage $A_t$ (where $A_t$ denotes $A_t(s_t, a_t)$):

$$A_t = \delta_t + \gamma\lambda A_{t+1}, \quad \delta_t = r_t + \gamma v(s_{t+1}) - v(s_t). \quad (16)$$

This advantage $A_t$ is then used in the standard PPO surrogate objective. The complete training procedure, which coordinates the element-wise distributional value updates and the risk-neutral policy optimization, is summarized in Algorithm 1 in Appendix C. Additionally, a summary comparison between Vanilla PPO and DisPPO is provided in Table 1.

*Table 1.* Comparison of Critic Implementation: Vanilla PPO vs. DisPPO

| Component | Vanilla PPO | DisPPO (Ours) |
|---|---|---|
| **Target** | Scalar $\mathbb{E}[Z]$ | Full Distribution $Z$ |
| **Operator** | Scalar $\mathcal{T}_V^{\pi,\lambda}$ | Dist. $\mathcal{T}^{\pi,\lambda}$ |
| **Output** | Scalar $v \in \mathbb{R}$ | $N$ Quantiles $\{q^{(i)}\}$ |
| **Advantage** | Scalar $A_t$ | Scalar $A_t$ (mean of quantiles) |
| **Loss** | MSE | Quantile Regression |

## 4. Theoretical Analysis

A fundamental requirement for the stability of the critic in actor-critic methods is that the value update operator acts as a contraction mapping. In this section, we analyze the convergence properties of our **distributional policy evaluation** mechanism. Specifically, we establish the contraction of the ideal distributional $\lambda$-operator, and then extend this result to the practical setting involving projection onto a finite number of quantiles. Note that this analysis pertains to the convergence of the critic for a fixed policy $\pi$, providing the theoretical foundation for the stability of the learning signal used in DisPPO.

### 4.1. Contraction of the $\lambda$-Operator

First, we consider the ideal case where the return distribution can be represented exactly.

**Theorem 4.1** (Contraction of $\mathcal{T}^{\pi,\lambda}$). *For any $\lambda \in [0, 1)$ and $\gamma \in [0, 1)$, the distributional $\lambda$-Bellman operator $\mathcal{T}^{\pi,\lambda}$ (Eq. 10) is a contraction mapping with respect to the maximal Wasserstein metric $\bar{d}_p$:*

$$\bar{d}_p(\mathcal{T}^{\pi,\lambda} Z_1, \mathcal{T}^{\pi,\lambda} Z_2) \leq \frac{\gamma(1-\lambda)}{1-\gamma\lambda} \bar{d}_p(Z_1, Z_2). \qquad (17)$$

This implies that repeated application of $\mathcal{T}^{\pi,\lambda}$ converges to a unique fixed-point distribution $Z^\pi$. We defer the proof to Appendix A.

### 4.2. Convergence of the Projected Operator

Our practical update rule iterates the composite operator $\Pi_{W_1} \mathcal{T}^{\pi,\lambda}$, applying the distributional $\lambda$-Bellman operator followed by a projection onto the parametric manifold $\mathcal{Z}_N$ defined in Section 2. While $\Pi_{W_1}$ minimizes the $W_1$ distance by definition (Eq. 5), establishing global convergence requires analyzing the interaction between the operator and the projection under a metric.

**Theorem 4.2** (Contraction of the Projected Operator). *Let $\bar{d}_\infty$ denote the maximal $\infty$-Wasserstein metric defined over the state space. For any two value distributions $Z_1, Z_2 \in \mathcal{Z}$,*

*the combined operator $\Pi_{W_1}\mathcal{T}^{\pi,\lambda}$ satisfies:*

$$\bar{d}_\infty(\Pi_{W_1}\mathcal{T}^{\pi,\lambda}Z_1, \Pi_{W_1}\mathcal{T}^{\pi,\lambda}Z_2) \leq \frac{\gamma(1-\lambda)}{1-\gamma\lambda}\bar{d}_\infty(Z_1, Z_2). \tag{18}$$

Theorem 4.2 provides the theoretical justification for DisPPO, guaranteeing that our $\lambda$-return mechanism converge to a fixed point when combined with the quantile-based updates. We defer the proof to Appendix A.

## 5. Related Work

**Reinforcement Learning for LLMs.** RL has become central to enhancing LLM reasoning, especially in long chain-of-thought (CoT) tasks (Wei et al., 2022; OpenAI, 2024; Guo et al., 2025; Shao et al., 2024). Value-model-free methods such as GRPO and DAPO avoid critics by redistributing trajectory-level rewards, offering stability and scalability for long sequences (Shao et al., 2024; Yu et al., 2025). In contrast, value-model-based methods like PPO provide finer credit assignment but often suffer collapse in long-CoT settings (Schulman et al., 2017; Yuan et al., 2025). Recent frameworks such as VAPO improve stability via value-regularization and length normalization (Yue et al., 2025), highlighting renewed interest in critic-based training. Orthogonal efforts address exploration and diversity in RLVR: count-based intrinsic rewards (Zhang et al., 2026) and divergence-aware training (Li et al., 2026a;b) tackle reward sparsity and mode collapse, while SQL-ASTRA (Anonymous, 2026) alleviates sparse feedback in agentic SQL tasks via trajectory aggregation.

**Distributional Reinforcement Learning.** The advent of Distributional Reinforcement Learning (DistRL) marked a paradigm shift from modeling the expected return to capturing the full distribution of stochastic outcomes (Bellemare et al., 2017; Qu et al., 2019). Early works in this domain were predominantly value-based and designed for discrete action spaces. The seminal C51 algorithm (Bellemare et al., 2017) pioneered this approach by parameterizing the return distribution with a discrete set of uniformly spaced atoms over a predefined range. This idea was subsequently advanced by QR-DQN (Dabney et al., 2018b), which proposed using a discrete set of quantiles, and further generalized by IQN (Dabney et al., 2018a), which learns the full continuous quantile function, demonstrating enhanced performance and flexibility over fixed-atom representations. The extension of distributional concepts to policy-gradient methods suitable for continuous control tasks yielded algorithms such as Reactor (Gruslys et al., 2018) and Distributed Distributional Deterministic Policy Gradients (D4PG) (Barth-Maron et al., 2018). SDPG (Singh et al., 2022) further introduced a sample-based distributional policy gradient method via reparameterization. However, these actor-critic variants are primarily designed for continuous control robotics tasks and rely heavily on off-policy mechanisms, such as extensive experience replay, to stabilize training. Maintaining the massive replay buffers required for such methods is memory-prohibitive for large-scale language models. Moreover, the reliance on stale off-policy data often introduces optimization instability compared to the on-policy updates favored in LLM fine-tuning (e.g., PPO). Furthermore, other adaptations of distributional critics often focus on risk-sensitive objectives (e.g., CVaR) (Tang et al., 2020; Singh et al., 2020), diverging from the standard expectation maximization objective required for reasoning accuracy.

## 6. Experiments

We evaluate **DisPPO** across diverse reasoning scenarios to investigate whether explicitly modeling the *full distribution* of state values enhances policy learning in large language models. While standard PPO relies on a scalar critic that compresses return information, DisPPO employs a distributional critic to estimate the full return distribution $Z_\pi(s)$. We design our experiments to probe this modeling choice in three distinct settings: single-shot mathematical reasoning, cross-domain structured programming (SQL), and multi-turn tool-augmented reasoning with delayed rewards.

Our evaluation is structured into three parts:

- **Mathematical Reasoning:** We assess performance on challenging math benchmarks to measure improvements in reasoning accuracy.

- **SQL Code Generation:** We examine cross-domain generalization on SQL generation, a task characterized by strict syntax constraints and sparse rewards.

- **Multi-turn Tool-augmented Reasoning:** We evaluate the effectiveness of our method in multi-turn interaction-based reasoning scenarios.

### 6.1. Experimental Setup

**Datasets and Models.** We utilize task-specific datasets and backbone models for each domain. For **mathematical reasoning**, we train on the DAPO-Math-17K dataset (Yu et al., 2025), comprising 17K problems from competitions and web corpora, using Qwen2.5-Math-7B (Yang et al., 2024) as the backbone. For **SQL code generation**, we train on the Bird training set (Li et al., 2023) using Llama-3.1-8B-Instruct (Grattafiori et al., 2024) to test reasoning in program-like environments. For **multi-turn tool-augmented reasoning**, we utilize the same DAPO-Math-17K dataset but adopt Qwen2.5-7B-Instruct (Yang et al., 2024), requiring the model to interact with tools over multiple turns.

*Table 2.* Performance on mathematical reasoning with pass@*k* (top) and avg@*k* (bottom). Best results in **bold**, second best underlined, within each backbone group.In all following tables, Δ rows denote the performance gain of DisPPO over PPO.

*(a)* avg@*k* results

| | AIME25 | AIME24 | Minerva | MATH500 | OlympiadBench | College | Avg. |
|---|---|---|---|---|---|---|---|
| | *avg@256* | *avg@256* | *avg@16* | *avg@16* | *avg@16* | *avg@8* | |
| *Qwen2.5-Math-7B* | 4.4 | 10.7 | 16.9 | 47.5 | 20.4 | 22.1 | 20.3 |
| + GRPO | 11.2 | 28.7 | 41.8 | 79.0 | 40.3 | 42.0 | 40.5 |
| + DAPO | 16.5 | 31.9 | 41.0 | 81.5 | 41.4 | 41.0 | 42.2 |
| + DAPO w/ Entropy Adv. | 17.2 | 33.3 | 44.5 | 80.9 | 41.4 | 41.6 | 43.2 |
| + PPO w/ Entropy Adv. | 17.1 | 31.9 | 45.0 | 81.0 | 42.9 | 43.0 | 43.4 |
| + PPO | 16.4 | 31.3 | 44.7 | 81.0 | 42.8 | 42.7 | 43.1 |
| + DisPPO(ours) | **19.2** | **33.7** | **46.9** | **82.5** | **44.2** | **43.5** | **45.0** |
| Δ | +2.8 | +2.4 | +2.2 | +1.5 | +1.4 | +0.8 | +1.9 |

*(b)* pass@*k* results

| | AIME25 | AIME24 | Minerva | MATH500 | OlympiadBench | College | Avg. |
|---|---|---|---|---|---|---|---|
| | *pass@256* | *pass@256* | *pass@16* | *pass@16* | *pass@16* | *pass@8* | |
| *Qwen2.5-Math-7B* | 53.3 | 70.0 | 50.4 | 88.6 | 56.7 | 44.2 | 60.5 |
| + GRPO | 50.0 | 76.7 | 64.0 | 91.8 | 59.7 | 49.2 | 65.2 |
| + DAPO | 56.7 | 76.7 | 66.9 | 92.0 | 60.9 | 48.3 | 66.9 |
| + DAPO w/ Entropy Adv. | 60.0 | 83.3 | 66.5 | 91.4 | 57.6 | 48.5 | 67.9 |
| + PPO w/ Entropy Adv. | 60.0 | 76.7 | 68.0 | 91.2 | 62.0 | 50.4 | 68.1 |
| + PPO | 60.0 | 73.3 | 67.7 | 91.0 | 61.3 | 50.2 | 67.3 |
| + DisPPO(ours) | **63.3** | **86.7** | **68.4** | **93.6** | **64.6** | **50.8** | **71.2** |
| Δ | +3.3 | +13.4 | +0.7 | +2.6 | +3.3 | +0.6 | +3.9 |

**Baselines and Configuration.** We compare DisPPO against three primary baselines: **GRPO** (Shao et al., 2024), **DAPO** (Yu et al., 2025), and standard **PPO** (Schulman et al., 2017). All actor-critic methods are implemented using the veRL framework (Sheng et al., 2025) to ensure a fair, unified training environment. To build strong baselines, we adopt several optimization techniques commonly used in recent large-scale RL for reasoning models, following DAPO and VAPO (Yue et al., 2025), including Clip-Higher, Token-level Loss, Critic Pretraining, and Group Sampling.

To ensure a rigorous evaluation against state-of-the-art standards, we additionally include **entropy-regularized variants** of PPO and DAPO (denoted as *PPO/DAPO + Entropy Adv*) (Cheng et al., 2025). While our work focuses on improving critic representation rather than exploration mechanisms, entropy-based advantage shaping is a widely adopted technique for enhancing exploration in reasoning tasks. Including these strong baselines allows us to verify whether the gains from DisPPO's distributional critic hold even against highly optimized training configurations.

For **multi-turn tool-augmented reasoning**, we align our setup with ReTool (Feng et al., 2025) for fair comparison,

evaluating PPO, DAPO, and their entropy-based variants without dynamic sampling.

For **DisPPO**, we simply replace the scalar critic head in PPO with a distributional head using $N = 51$ quantiles. All other hyperparameters remain consistent with standard PPO. Detailed hyperparameters are provided in Appendix F.2.

**Computational Efficiency.** DisPPO is computationally efficient. It adds only a small training-time overhead of about 4% compared to vanilla PPO, and it adds no overhead at inference time. A detailed analysis of compute cost and training overhead is given in Appendix L.

**Benchmarks and Metrics.** We follow the rigorous evaluation protocol of Cheng et al. (2025). **Math benchmarks** include AIME2024/2025 (MAA, 2025), MATH500 (Hendrycks et al., 2021), AMC (of America, 2023), OlympiadBench (He et al., 2024), College Math (Tang et al., 2024), and Minerva (Lewkowycz et al., 2022). **Tool-augmented benchmarks** include AIME2024/2025 and the challenging BeyondAIME (Seed et al., 2025). **SQL benchmarks** include the Bird and Spider test sets (Li et al., 2023; Yu et al., 2019).

We report unbiased estimates for **pass@$k$** (probability of at least one correct solution) and **avg@$k$** (expected correctness). We prioritize avg@$k$ as it serves as a more direct proxy for the quality of the learned policy.

## 6.2. Main Results

**Mathematical Reasoning.** Table 2 reports results on mathematical reasoning benchmarks. DisPPO consistently improves avg@$k$ compared to PPO, increasing the overall average from 43.1 to 45.0 (+1.9). The gains are observed across all benchmarks, with larger improvements on more challenging datasets such as AIME24/25, OlympiadBench, and Minerva. To further verify robustness against a strong value-regularized baseline, Appendix H.1 reports three-seed mean±std comparisons among PPO, VAPO, and DisPPO, where DisPPO remains best on average.

**SQL Code Generation.** Results on SQL code generation are summarized in Table 3. DisPPO improves greedy accuracy over PPO on both in-domain Bird (+2.1) and out-of-domain Spider (+4.2). The larger gain on Spider suggests improved robustness under domain shift.

**Multi-turn Tool-Augmented Reasoning.** On multi-turn reasoning benchmarks (Table 5, avg@64), DisPPO achieves the best performance among the compared methods. Compared with PPO, DisPPO improves the average score from 29.1 to 30.2 (+1.1), indicating improved performance in delayed-reward, long-horizon reasoning settings.

**Standard RLHF with Reward Model.** To evaluate whether DisPPO generalizes beyond verifiable reasoning tasks with binary rewards, we further conduct experiments in a standard reward-model-based RLHF setup. We use LLaMA-3-8B (Grattafiori et al., 2024) as the backbone and follow the open-source PPO RLHF pipeline of Yin et al. (2025) on UltraFeedback preference data (Cui et al., 2024), where a learned reward model assigns a continuous scalar reward only at EOS. For a controlled comparison, we keep the backbone model, reward model, dataset, and PPO training recipe unchanged, and replace only the scalar critic in PPO with a distributional critic. We additionally include VAPO (Yue et al., 2025) as a baseline. As shown in Table 4, DisPPO improves over PPO and VAPO across AlpacaEval 2 (Dubois et al., 2024), Arena-Hard (Li et al., 2024), and MT-Bench (Zheng et al., 2023).

## 6.3. Analysis: Value Prediction Accuracy

A core hypothesis of this work is that *distributional supervision leads to more accurate mean value estimation* than direct scalar regression. To validate this, we compare the critic's predicted values against ground-truth Monte Carlo returns computed from rollout trajectories on AIME25.

Table 6 reports the Mean Absolute Error between the predicted value and the ground truth at various prefix positions. DisPPO consistently achieves lower MAE compared to the scalar baseline across both absolute and relative token positions. **This empirical finding corroborates our intuition:** although the final policy update only utilizes the mean, the auxiliary task of modeling the full distribution forces the network to capture the underlying reasoning uncertainty. This mitigates the information collapse inherent in MSE loss, resulting in a significantly more accurate value estimator. Table 7 further provides a finer-grained bin analysis by Monte Carlo target region. DisPPO's advantage is concentrated in the dominant boundary regions $[0, 0.3)$ and $[0.7, 1.0]$, which account for 94.1% of all prefix states.

## 6.4. Ablation Studies

We run ablation studies to analyze the design choices of DisPPO. We focus on the critic architecture, the training loss, and the number of quantiles used in the value model.

First, we study the critic design for mathematical reasoning in the Appendix H.2 Table 14 reports results for different critic losses and architectures on AIME24 and AIME25, with the center–delta quantile critic performing best overall. In addition, we conduct a controlled comparison between DisPPO and a parametric distributional critic (GMAC) to further examine the impact of distribution parameterization in the Appendix K. Then, we examine the effect of the number of quantiles on SQL generation in the Appendix H.3 Results on BIRD and Spider are summarized in Table 15.

## 7. Conclusion

We presented DisPPO, a distributional actor–critic framework that integrates a quantile-based value distribution into on-policy PPO for large language model reasoning, with theoretical convergence guarantees under the Wasserstein metric. Empirically, DisPPO consistently outperforms PPO and value-free baselines across mathematical reasoning, SQL code generation, multi-turn tool-augmented reasoning, and standard RLHF, while introducing only modest computational overhead. Our analysis reveals that the distributional objective leads to more accurate mean value estimates than direct scalar regression, with particular improvements in the dominant boundary regions of the return distribution. These findings suggest that distributional value modeling is a promising direction for improving critic-based reinforcement learning in large language models. A current limitation is that our evaluation focuses on models up to 8B parameters; scaling to larger models and exploring risk-sensitive policy objectives with the learned return distribution are interesting avenues for future work.

*Table 3.* Performance on SQL code generation with greedy sampling and pass@$k$. **Greedy** denotes single-sample decoding using greedy token selection without sampling. Best results in **bold**, second best underlined, within each backbone group.

| Model | **Bird** (in domain) | | | **Spider** (out of domain) | | | Avg |
| --- | --- | --- | --- | --- | --- | --- | --- |
| | greedy | *pass@8* | *pass@16* | greedy | *pass@8* | *pass@16* | |
| *Llama-3.1-8B-Instruct* | 42.4 | 68.5 | 75.1 | 69.0 | **91.0** | **94.6** | 73.4 |
| + GRPO | 60.7 | 72.2 | 74.6 | 74.7 | 81.0 | 82.9 | 74.4 |
| + DAPO | 63.2 | 73.9 | 75.9 | 76.8 | 86.1 | 87.2 | 77.2 |
| + DAPO w/ Entropy Adv. | 62.3 | 73.2 | 75.9 | 77.5 | 86.1 | 87.6 | 77.1 |
| + PPO w/ Entropy Adv. | 62.0 | 73.3 | 76.0 | 77.6 | 86.3 | 87.9 | 77.2 |
| + PPO | 61.4 | 73.1 | 74.9 | 77.0 | 85.2 | 86.6 | 76.3 |
| + DisPPO (ours) | **63.5** | **75.0** | **77.4** | **81.2** | 87.5 | 88.9 | **78.9** |
| Δ | +2.1 | +1.9 | +2.5 | +4.2 | +2.3 | +2.3 | +2.6 |

*Table 4.* Standard RLHF comparison on LLaMA-3-8B trained with UltraFeedback. Best results are in **bold** and second best are underlined.

| Method | AlpacaEval 2 (WR) | Arena-Hard | MT-Bench |
| --- | --- | --- | --- |
| VAPO | 23.78 | 18.5 | 7.23 |
| PPO | 20.77 | 18.3 | 7.18 |
| DisPPO (ours) | **30.14** | **19.3** | **7.41** |
| Δ vs. PPO | +9.37 | +1.0 | +0.23 |

*Table 5.* Performance on multi-turn tool-augmented reasoning benchmarks (avg@64). Best results in **bold**, second best underlined, within each backbone group.

| Model | AIME25 | AIME24 | BeyondAIME | Avg |
| --- | --- | --- | --- | --- |
| *Qwen2.5-7b-instruct* | 8.0 | 14.3 | 6.5 | 9.6 |
| + DAPO | 29.4 | 39.7 | 13.8 | 27.6 |
| + DAPO w/ Entropy Adv. | 31.2 | 40.9 | 14.5 | 28.9 |
| + PPO w/ Entropy Adv. | 31.0 | 43.5 | 14.3 | 29.6 |
| + PPO | 29.2 | 44.1 | 14.0 | 29.1 |
| + DisPPO (ours) | **31.4** | **44.4** | **14.7** | **30.2** |
| Δ | +2.2 | +0.3 | +0.7 | +1.1 |

*Table 6.* Value prediction error (MAE) against prefix-based Monte Carlo targets on AIME25 (30 problems; 64 rollouts per problem; 256 continuations per prefix). We report results for both a fixed prefix position (500 tokens) and relative prefix positions (25%, 50%, and 75% of the generated sequence length). Lower is better.

| Method | 500 tokens | 25% | 50% | 75% | Avg |
| --- | --- | --- | --- | --- | --- |
| PPO | 0.3352 | 0.2923 | 0.2865 | 0.2925 | 0.3016 |
| DisPPO (ours) | **0.3044** | **0.2654** | **0.2703** | **0.2598** | **0.2749** |
| Δ | -0.0308 | -0.0269 | -0.0162 | -0.0327 | -0.0267 |

*Table 7.* Value prediction MAE stratified by Monte Carlo target bin on AIME25. DisPPO achieves lower MAE in the dominant boundary regions $[0, 0.3]$ and $[0.7, 1.0]$, while PPO is better only in the narrow middle region $[0.3, 0.7)$.

| MC target bin | Share | DisPPO MAE | PPO MAE | Diff (D−P) |
| --- | --- | --- | --- | --- |
| $[0.0, 0.3)$ | 74.5% | **0.3242** | 0.3446 | −0.020 |
| $[0.3, 0.7)$ | 5.9% | 0.2346 | **0.1723** | +0.062 |
| $[0.7, 1.0]$ | 19.6% | **0.1573** | 0.1714 | −0.014 |

# Acknowledgements

This work was supported by the National Natural Science Foundation of China (Grant No. 82394432 and 92249302) and the Shanghai Municipal Science and Technology Major Project (Grant No. 2023SHZDZX02).

# Impact Statement

This work improves the value estimation mechanism in reinforcement learning for large language models, leading to more accurate reasoning capabilities. On the positive side, stronger reasoning models can benefit scientific discovery, education, and automated problem-solving. However, improved reasoning may also lower the barrier for generating persuasive misinformation or assisting in dual-use applications. Additionally, RL-based post-training of LLMs requires substantial computational resources, contributing to energy consumption and associated environmental costs. We encourage the community to pair advances in LLM reasoning with robust safety evaluations, alignment research, and responsible deployment practices.

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

# A. Proof

In this section, we provide the proof of Theorem 4.1 and 4.2. Before proceeding, we clarify a notational convention used throughout this appendix: **State vs. State-Action Value:** While our proposed method (DisPPO) estimates the state-value distribution $Z(s)$, the theoretical foundations of distributional RL are canonically established on state-action value distributions $Z(s, a)$ (Bellemare et al., 2017). Since the contraction properties of the Bellman operator are preserved regardless of this distinction (as $Z(s)$ essentially aggregates $Z(s, a)$ over the policy $\pi$), we formulate our proofs using the standard notation $Z(s, a)$ to align with established literature. The results apply analogously to $Z(s)$ without loss of generality.

## A.1. Contraction of the distributional $\lambda$-Bellman operator

We start with the contraction property of the vanilla distributional Bellman operator, where Lemma A.1 is adapted from (Bellemare et al., 2017). We include it here for completeness and extend it to the general $n$-step case. Following the convention in (Bellemare et al., 2017), we view the reward function as a random vector $R \in \mathcal{Z}$ and define the transition operator $P^\pi : \mathcal{Z} \to \mathcal{Z}$ as:

$$P^\pi Z(s, a) :\overset{D}{=} Z(S', A'), \tag{19}$$
$$S' \sim P(\cdot \mid s, a), \quad A' \sim \pi(\cdot \mid S'),$$

where we use capital letters to emphasize the random nature of the next state-action pair $(S', A')$. We define the distributional Bellman operator $\mathcal{T}^\pi : \mathcal{Z} \to \mathcal{Z}$ as:

$$\mathcal{T}^\pi Z(s, a) :\overset{D}{=} R(s, a) + \gamma P^\pi Z(s, a). \tag{20}$$

**Lemma A.1.** *Let $Z_1, Z_2 \in \mathcal{Z}$ be two value distributions. Let $d_p$ denote the $p$-Wasserstein distance between two distributions, and define the metric over the value function space as $\bar{d}_p(Z_1, Z_2) := \sup_{s,a} d_p(Z_1(s, a), Z_2(s, a))$.*

*The operator $\mathcal{T}^\pi$ is a $\gamma$-contraction in $\bar{d}_p$. Specifically, :*

$$\bar{d}_p(\mathcal{T}^\pi Z_1, \mathcal{T}^\pi Z_2) \leq \gamma \bar{d}_p(Z_1, Z_2). \tag{21}$$

*Furthermore, by induction, for any $n \geq 1$, the $n$-step operator satisfies:*

$$\bar{d}_p((\mathcal{T}^\pi)^n Z_1, (\mathcal{T}^\pi)^n Z_2) \leq \gamma^n \bar{d}_p(Z_1, Z_2). \tag{22}$$

*Proof.* The proof relies on the properties of the Wasserstein metric, specifically shift-invariance ($d_p(A + X, A + Y) = d_p(X, Y)$) and absolute homogeneity of degree 1 ($d_p(cX, cY) = |c| d_p(X, Y)$).

**1. Base Case (One-step contraction):** By the properties of $d_p$, for any $(s, a)$, we have:

$$
\begin{aligned}
d_p(\mathcal{T}^\pi Z_1(s, a), \mathcal{T}^\pi Z_2(s, a)) &= d_p(R(s, a) + \gamma P^\pi Z_1(s, a), R(s, a) + \gamma P^\pi Z_2(s, a)) \\
&= d_p(\gamma P^\pi Z_1(s, a), \gamma P^\pi Z_2(s, a)) \quad \text{(Shift-invariance)} \\
&= \gamma d_p(P^\pi Z_1(s, a), P^\pi Z_2(s, a)). \quad \text{(Homogeneity)}
\end{aligned}
$$

By the definition of the transition operator $P^\pi$, $P^\pi Z(s, a)$ is distributed as $Z(S', A')$. The distance between these mixtures is bounded by the supremum distance over the support of $(S', A')$:

$$
\begin{aligned}
\gamma d_p(P^\pi Z_1(s, a), P^\pi Z_2(s, a)) &\leq \gamma \sup_{s', a'} d_p(Z_1(s', a'), Z_2(s', a')) \\
&= \gamma \bar{d}_p(Z_1, Z_2).
\end{aligned}
$$

Taking the supremum over $(s, a)$ on both sides yields:

$$\bar{d}_p(\mathcal{T}^\pi Z_1, \mathcal{T}^\pi Z_2) \leq \gamma \bar{d}_p(Z_1, Z_2). \tag{23}$$

**2. Inductive Step:** We proceed by induction on $n$. The base case ($n = 1$) is shown in Eq. (23). Assume the hypothesis holds for $n$, i.e., $\bar{d}_p((\mathcal{T}^\pi)^n Z_1, (\mathcal{T}^\pi)^n Z_2) \leq \gamma^n \bar{d}_p(Z_1, Z_2)$.

Consider the case for $n + 1$. Let $Z_1' = (\mathcal{T}^\pi)^n Z_1$ and $Z_2' = (\mathcal{T}^\pi)^n Z_2$. Applying the base case contraction to $Z_1'$ and $Z_2'$:

$$
\begin{aligned}
\bar{d}_p \left( (\mathcal{T}^\pi)^{n+1} Z_1, (\mathcal{T}^\pi)^{n+1} Z_2 \right) &= \bar{d}_p \left( \mathcal{T}^\pi Z_1', \mathcal{T}^\pi Z_2' \right) \\
&\leq \gamma \bar{d}_p(Z_1', Z_2') \quad \text{(using Eq. 23)} \\
&= \gamma \bar{d}_p \left( (\mathcal{T}^\pi)^n Z_1, (\mathcal{T}^\pi)^n Z_2 \right) \\
&\leq \gamma \cdot \gamma^n \bar{d}_p(Z_1, Z_2) \quad \text{(using inductive hypothesis)} \\
&= \gamma^{n+1} \bar{d}_p(Z_1, Z_2).
\end{aligned}
$$

This concludes the proof. $\qquad \square$

Using Lemma A.1, we now prove the contraction property of the generalized operator $\mathcal{T}^{\pi,\lambda}$, i.e., Theorem 4.1.

*Proof.* Let $Z_1, Z_2 \in \mathcal{Z}$ be two arbitrary value distributions. We analyze the Wasserstein distance at a fixed state-action pair $(s, a)$. By the convexity of the Wasserstein metric, we have:

$$
d_p \left( \mathcal{T}^{\pi,\lambda} Z_1(s,a), \mathcal{T}^{\pi,\lambda} Z_2(s,a) \right) \leq (1 - \lambda) \sum_{n=0}^{\infty} \lambda^n d_p \left( (\mathcal{T}^\pi)^{n+1} Z_1(s,a), (\mathcal{T}^\pi)^{n+1} Z_2(s,a) \right). \tag{24}
$$

We now apply Lemma A.1 to bound each term in the summation. Specifically, we substitute the local distance bound $d_p((\mathcal{T}^\pi)^{n+1} Z_1(s,a), \dots) \leq \gamma^{n+1} \bar{d}_p(Z_1, Z_2)$:

$$
d_p \left( \mathcal{T}^{\pi,\lambda} Z_1(s,a), \mathcal{T}^{\pi,\lambda} Z_2(s,a) \right) \leq (1 - \lambda) \sum_{n=0}^{\infty} \lambda^n \left[ \gamma^{n+1} \bar{d}_p(Z_1, Z_2) \right] \tag{25}
$$

$$
= (1 - \lambda) \gamma \left( \sum_{n=0}^{\infty} (\lambda \gamma)^n \right) \bar{d}_p(Z_1, Z_2). \tag{26}
$$

The geometric series converges to $\frac{1}{1 - \lambda \gamma}$, yielding:

$$
d_p \left( \mathcal{T}^{\pi,\lambda} Z_1(s,a), \mathcal{T}^{\pi,\lambda} Z_2(s,a) \right) \leq \frac{\gamma(1 - \lambda)}{1 - \gamma \lambda} \bar{d}_p(Z_1, Z_2). \tag{27}
$$

Since this inequality holds for any arbitrarily chosen $(s, a)$, taking the supremum over all state-action pairs confirms the global contraction:

$$
\bar{d}_p(\mathcal{T}^{\pi,\lambda} Z_1, \mathcal{T}^{\pi,\lambda} Z_2) \leq \frac{\gamma(1 - \lambda)}{1 - \gamma \lambda} \bar{d}_p(Z_1, Z_2). \tag{28}
$$

$\qquad \square$

### A.2. Combining Bellman update with the projection

Let $Y$ be a distribution with a bounded first moment and $U \in \mathcal{Z}_Q$ be a uniform distribution supported on $\{q_1, \dots, q_N\}$ with $q_1 \leq \dots \leq q_N$. The 1-Wasserstein distance between them can be explicitly written as:

$$
W_1(Y, U) = \sum_{i=1}^{N} \int_{\tau_{i-1}}^{\tau_i} |F_Y^{-1}(\omega) - q_i| d\omega, \tag{29}
$$

where $\tau_i = \frac{i}{N}$ for $i = 0, \dots, N$ are the quantile levels. Following the similar proof strategy (Bellemare et al., 2017), we have the following lemma.

**Lemma A.2.** *Let $\nu_1, \nu_2$ be any two probability distributions over the real numbers. Let $\Pi_{W_1}$ be the projection operator onto $\mathcal{Z}_Q$ (distributions with support size $N$). Then, the $\infty$-Wasserstein distance between the projected distributions satisfies:*

$$
W_\infty(\Pi_{W_1} \nu_1, \Pi_{W_1} \nu_2) = \max_{i=1,\dots,N} \left| F_{\nu_1}^{-1} \left( \frac{2i - 1}{2N} \right) - F_{\nu_2}^{-1} \left( \frac{2i - 1}{2N} \right) \right|. \tag{30}
$$

*Proof.* The minimization of $W_1(Z, Z_\theta)$ decomposes into $N$ independent minimization problems. For each $i \in \{1, \ldots, N\}$, we minimize the objective function $L_i(\theta) = \int_{\tau_{i-1}}^{\tau_i} |F^{-1}(\omega) - \theta| d\omega$.

The integrand $g(\theta, \omega) = |F^{-1}(\omega) - \theta|$ is convex in $\theta$. Its subgradient with respect to $\theta$ is:

$$\partial_\theta g(\theta, \omega) = \begin{cases} -1 & \text{if } \theta < F^{-1}(\omega) \\ [-1, 1] & \text{if } \theta = F^{-1}(\omega) \\ 1 & \text{if } \theta > F^{-1}(\omega). \end{cases}$$

By Leibniz's integral rule, the subgradient of the objective $L_i(\theta)$ is:

$$\partial L_i(\theta) = \int_{\tau_{i-1}}^{\tau_i} \partial_\theta g(\theta, \omega) d\omega = \int_{\tau_{i-1}}^{F(\theta)} (1) d\omega + \int_{F(\theta)}^{\tau_i} (-1) d\omega,$$

assuming $F(\theta) \in [\tau_{i-1}, \tau_i]$. Evaluating the integrals yields:

$$\partial L_i(\theta) = (F(\theta) - \tau_{i-1}) - (\tau_i - F(\theta)) = 2F(\theta) - (\tau_{i-1} + \tau_i).$$

Setting the subgradient to 0 gives the optimality condition $F(\theta) = \frac{\tau_{i-1} + \tau_i}{2}$. Since $\tau_i = i/N$, the midpoint is $\frac{2i-1}{2N}$. Thus, the minimizer is the quantile at the midpoint of the interval:

$$\theta_i^* = F^{-1} \left( \frac{2i-1}{2N} \right).$$

Consequently, the projections are given by:

$$\Pi_{W_1} \nu_1 = \frac{1}{N} \sum_{i=1}^{N} \delta_{q_{1,i}}, \quad \Pi_{W_1} \nu_2 = \frac{1}{N} \sum_{i=1}^{N} \delta_{q_{2,i}},$$

where $q_{k,i} = F_{\nu_k}^{-1}(\frac{2i-1}{2N})$.

Recall that for two distributions $Y, U$ supported on $N$ sorted atoms, the $\infty$-Wasserstein distance is given by $W_\infty(Y, U) = \max_i |y_i - u_i|$. Applying this to our projections:

$$W_\infty(\Pi_{W_1} \nu_1, \Pi_{W_1} \nu_2) = \max_{i=1,\ldots,N} |q_{1,i} - q_{2,i}|.$$

Substituting the expressions for $q_{1,i}$ and $q_{2,i}$ concludes the proof. $\square$

In the following lemma, we extend the result on $\mathcal{T}^\pi$ in (Dabney et al., 2018b) to the case of $(\mathcal{T}^\pi)^k$.

**Lemma A.3.** *Consider an MDP with countable state and action spaces. Let $Z, Y$ be value distributions such that each state-action distribution $Z(s, a), Y(s, a)$ is given by a single Dirac. Consider the particular case where rewards are identically 0 and $\gamma = 1$, and let $\tau \in [0, 1]$. Denote by $\Pi_\tau$ the projection operator that maps a probability distribution onto a Dirac delta located at its $\tau$-th quantile. Then:*

$$\bar{d}_\infty(\Pi_\tau (\mathcal{T}^\pi)^k Z, \Pi_\tau (\mathcal{T}^\pi)^k Y) \leq \bar{d}_\infty(Z, Y).$$

*Proof.* Let $Z(s, a) = \delta_{\theta(s,a)}$ and $Y(s, a) = \delta_{\psi(s,a)}$ for each state-action pair $(s, a) \in \mathcal{S} \times \mathcal{A}$. Let $((s_i, a_i))_{i \in I_k}$ be the set of state-action pairs accessible from a starting pair $(s, a)$ in $k$ transitions, indexed by $I_k$, with transition probabilities $p_i$. We denote $\theta_i := \theta(s_i, a_i)$ and $\psi_i := \psi(s_i, a_i)$.

The Bellman updates yield mixture distributions with support $\{\theta_i\}_{i \in I_k}$ and $\{\psi_i\}_{i \in I_k}$. Let $\theta_u$ be the $\tau$-th quantile of the updated $Z$ distribution (for some $u \in I_k$), and $\psi_v$ be the $\tau$-th quantile of the updated $Y$ distribution (for some $v \in I_k$).

We aim to show that $|\theta_u - \psi_v| \leq \epsilon$, where $\epsilon = \bar{d}_\infty(Z, Y) = \sup_{i \in I_k} |\theta_i - \psi_i|$.

We proceed by contradiction. Assume without loss of generality that $\theta_u \leq \psi_v$. Suppose, for the sake of contradiction, that $|\theta_u - \psi_v| > \epsilon$. This implies:

$$\psi_v - \theta_u > \epsilon \implies \theta_u + \epsilon < \psi_v. \tag{31}$$

We introduce the following partitions of the indexing set $I_k$:

$$I_{\leq \theta_u} = \{i \in I_k \mid \theta_i \leq \theta_u\}, \qquad\qquad I_{>\theta_u} = \{i \in I_k \mid \theta_i > \theta_u\},$$
$$I_{<\psi_v} = \{i \in I_k \mid \psi_i < \psi_v\}, \qquad\qquad I_{\geq \psi_v} = \{i \in I_k \mid \psi_i \geq \psi_v\}.$$

We observe the disjoint unions $I_k = I_{\leq \theta_u} \cup I_{>\theta_u}$ and $I_k = I_{<\psi_v} \cup I_{\geq \psi_v}$.

If (31) holds, we must have $I_{\leq \theta_u} \cap I_{\geq \psi_v} = \emptyset$. To see this, consider any index $j \in I_{\leq \theta_u}$. We have $\theta_j \leq \theta_u$. By the definition of $\epsilon$, we know $\psi_j \leq \theta_j + \epsilon$. Combining these gives:

$$\psi_j \leq \theta_j + \epsilon \leq \theta_u + \epsilon < \psi_v.$$

Thus, $j \in I_{<\psi_v}$. This proves the subset inclusion $I_{\leq \theta_u} \subseteq I_{<\psi_v}$.

Since $\theta_u$ is the $\tau$-th quantile of $(\mathcal{T}^\pi)^k Z(s,a)$, we must have:

$$\sum_{i \in I_{\leq \theta_u}} p_i \geq \tau.$$

Due to the inclusion $I_{\leq \theta_u} \subseteq I_{<\psi_v}$, it follows that:

$$\sum_{i \in I_{<\psi_v}} p_i \geq \sum_{i \in I_{\leq \theta_u}} p_i \geq \tau.$$

This implies that the cumulative probability mass for values strictly less than $\psi_v$ is at least $\tau$. By definition, this means the $\tau$-th quantile of $(\mathcal{T}^\pi)^k Y(s,a)$ must be strictly less than $\psi_v$, which contradicts the fact that $\psi_v$ is defined as the $\tau$-th quantile.

Therefore, the assumption must be false, and we conclude that $|\theta_u - \psi_v| \leq \epsilon$, completing the proof. $\qquad\square$

Now we are in the position to prove the our main Theorem 4.2.

*Proof.* We aim to bound the distance $\bar{d}_\infty(\Pi_{W_1} \mathcal{T}^{\pi,\lambda} Z_1, \Pi_{W_1} \mathcal{T}^{\pi,\lambda} Z_2)$. Since the Wasserstein distance is invariant under translation, we can simplify the problem without loss of generality by assuming the reward function is identically zero, $r(s,a) = 0$ for all $(s,a)$. In this setting, the Bellman operator acts purely as a discount scaling on the value distributions.

Step 1: Reduction to Single Diracs via Expanded MDP

The value distributions $Z(s,a)$ are represented by uniform distributions over $N$ quantiles. Let $Z_1(s,a) = \frac{1}{N} \sum_{j=1}^N \delta_{\theta_j(s,a)}$ and $Z_2(s,a) = \frac{1}{N} \sum_{j=1}^N \delta_{\psi_j(s,a)}$, where $\theta_j$ and $\psi_j$ are the sorted atoms.

We construct an augmented MDP $\widetilde{\mathcal{M}}$ to reduce the analysis to distributions consisting of single Diracs. The state space of $\widetilde{\mathcal{M}}$ is $\mathcal{S} \times \{1, \ldots, N\}$. For any transition $(s,a) \to s'$ in the original MDP with probability $P(s'|s,a)$, we define transitions in $\widetilde{\mathcal{M}}$ from $(s,j)$ to a set of states $\{(s',m)\}_{m=1}^N$ with transition probabilities distributed uniformly, i.e., reaching $(s',m)$ with probability $P(s'|s,a)/N$.

We define the lifted value distributions $\widetilde{Z}_1, \widetilde{Z}_2$ on this augmented space as single Diracs:

$$\widetilde{Z}_1((s,j),a) = \delta_{\theta_j(s,a)}, \quad \widetilde{Z}_2((s,j),a) = \delta_{\psi_j(s,a)}.$$

By construction, the stochasticity of the original quantile mixture is now encoded in the transition dynamics of $\widetilde{\mathcal{M}}$. The maximal $\infty$-Wasserstein distance in the original MDP corresponds to the supremum distance in the augmented MDP:

$$\bar{d}_\infty(Z_1, Z_2) = \sup_{s,a} \max_j |\theta_j(s,a) - \psi_j(s,a)| = \sup_{\tilde{s},\tilde{a}} |\theta(\tilde{s},\tilde{a}) - \psi(\tilde{s},\tilde{a})|. \qquad (32)$$

Step 2: Contraction of the $k$-step Operator

Consider the $k$-step Bellman operator $(\mathcal{T}^\pi)^{k+1}$. In the zero-reward setting, applying this operator scales the support of the distribution by $\gamma^{k+1}$.

Applying Lemma A.2, the distance between projected distributions is determined by the maximum difference between their quantiles. Furthermore, Lemma A.3 establishes that for single Dirac value distributions, the projection of the Bellman update does not expand the distance relative to the scale of the update. Specifically, adapting Lemma A.3 to include the discount factor $\gamma$:

$$\bar{d}_\infty \left( \Pi_{W_1}(\mathcal{T}^\pi)^{k+1} Z_1, \Pi_{W_1}(\mathcal{T}^\pi)^{k+1} Z_2 \right) \le \gamma^{k+1} \bar{d}_\infty(Z_1, Z_2).$$

Step 3: Aggregation for the $\lambda$-Operator

Recall the definition of the distributional $\lambda$-operator:

$$\mathcal{T}^{\pi,\lambda} Z(s,a) := (1-\lambda) \sum_{k=0}^\infty \lambda^k (\mathcal{T}^\pi)^{k+1} Z(s,a).$$

The $\infty$-Wasserstein distance measures the horizontal shift between CDFs. Since $\mathcal{T}^{\pi,\lambda}$ is a convex combination of operators with contraction factors $\gamma^{k+1}$, and the projection $\Pi_{W_1}$ acts element-wise on the quantile levels (which is linear with respect to pure shifts in the value domain), the total contraction factor is the sum of the discounted weights:

$$\alpha = (1-\lambda) \sum_{k=0}^\infty \lambda^k \gamma^{k+1} = \gamma(1-\lambda) \sum_{k=0}^\infty (\gamma\lambda)^k = \frac{\gamma(1-\lambda)}{1-\gamma\lambda}.$$

Therefore,

$$\bar{d}_\infty(\Pi_{W_1} \mathcal{T}^{\pi,\lambda} Z_1, \Pi_{W_1} \mathcal{T}^{\pi,\lambda} Z_2) \le \alpha \bar{d}_\infty(Z_1, Z_2).$$

This completes the proof. $\qquad\square$

# B. Derivation of the Distributional Critic Objective

## B.1. Population Objective

Let $p_\pi$ denote the trajectory distribution induced by policy $\pi$ and environment dynamics, and let $d_\pi(s)$ be the corresponding stationary state distribution. For each state $s$, let $Z_\pi(s)$ denote the return distribution induced by the distributional GAE construction.

Denote by $F_{Z_\pi(s)}^{-1}(\tau)$ the quantile function of $Z_\pi(s)$ at level $\tau \in (0,1)$. The critic approximates this quantile function at fixed quantile levels $\{\tau_i\}_{i=1}^N$ via parametric outputs $\{q_\theta^{(i)}(s)\}_{i=1}^N$.

The population distributional quantile regression objective is

$$\mathcal{L}_{\text{pop}}(\theta) = \mathbb{E}_{s \sim d_\pi} \left[ \int_0^1 \mathbb{E}_{Z \sim Z_\pi(s)} \left[ \rho_\tau^\kappa \left( Z - Q_\theta^{-1}(\tau \mid s) \right) \right] d\tau \right], \tag{33}$$

where $\rho_\tau^\kappa(\cdot)$ denotes the quantile Huber loss.

## B.2. Monte Carlo Approximation

We approximate the integral over $\tau$ using a discrete quantile grid,

$$\int_0^1 f(\tau) \, d\tau \approx \frac{1}{N} \sum_{i=1}^N f(\tau_i), \tag{34}$$

and the expectation over $Z_\pi(s_t)$ using $N$ Monte Carlo samples $\{\hat{y}_t^{(j)}\}_{j=1}^N$.

Substituting

$$Q_\theta^{-1}(\tau_i \mid s_t) = q_\theta^{(i)}(s_t), \tag{35}$$

yields the per-state empirical loss

$$\ell(\theta; s_t) = \frac{1}{N^2} \sum_{i=1}^N \sum_{j=1}^N \rho_{\tau_i}^\kappa \left( \hat{y}_t^{(j)} - q_\theta^{(i)}(s_t) \right). \tag{36}$$

### B.3. Empirical Trajectory Objective

Since $d_\pi$ is unknown, the expectation over states is approximated using time averages along trajectories sampled from $\pi$. Given a trajectory $\tau = (s_0, a_0, r_0, \ldots, s_T)$, we obtain

$$\mathcal{L}_{\text{emp}}(\theta) = \mathbb{E}_{\tau \sim \pi}\left[\frac{1}{T}\sum_{t=0}^{T-1}\ell(\theta; s_t)\right]. \tag{37}$$

Substituting Eq. (36) yields

$$\mathcal{L}_{\text{emp}}(\theta) = \mathbb{E}_{\tau \sim \pi}\left[\frac{1}{T}\sum_{t=0}^{T-1}\frac{1}{N^2}\sum_{i=1}^{N}\sum_{j=1}^{N}\rho_{\tau_i}^\kappa\left(\hat{y}_t^{(j)} - q_\theta^{(i)}(s_t)\right)\right], \tag{38}$$

which corresponds exactly to 14 in the main paper.

**Remark.** Eq. (38) is a Monte Carlo approximation of the population objective in Eq. (33), obtained by discretizing the quantile integral, sampling the return distribution, and replacing the stationary state expectation with empirical trajectory averages.

## C. Algorithmic Summary of DisPPO Training

Algorithm 1 summarizes the complete DisPPO training loop, highlighting how the distributional critic is updated with D-GAE while the actor remains a standard PPO update using scalar advantages. Note that the actor update is identical to standard PPO; all modifications are confined to the critic.

---

**Algorithm 1** DisPPO: Distributional PPO with D-GAE

---

**Require:** Policy $\pi_\psi$, distributional critic $q_\theta$, dataset $\mathcal{D}$, horizon $T$, quantile count $N$, discount $\gamma$, GAE parameter $\lambda$, PPO clip $\epsilon$.

1: **for** iteration $= 1, 2, \ldots$ **do**
2:     **Rollout:** Sample prompts $x \sim \mathcal{D}$ and generate trajectories $\tau = \{(s_t, a_t, r_t, s_{t+1})\}_{t=0}^{T-1}$ under $\pi_\psi$.
3:     **Critic forward pass:** For each $s_t$, compute ordered quantiles $\{q_\theta^{(i)}(s_t)\}_{i=1}^{N}$.
4:     **Distributional TD errors:** Compute per-quantile TD errors $\delta_t^{(i)}$ according to Eq. (11).
5:     **Distributional GAE (D-GAE):** Accumulate per-quantile distributional advantages $\mathcal{A}_t^{(i)}$ via the backward recursion in Eq. (12).
6:     **Distributional targets:** Construct multi-step regression targets $\hat{y}_t^{(i)}$ using Eq. (13).
7:     **Critic update:** Update $\theta$ by minimizing the quantile Huber loss in Eq. (14).
8:     **Scalar value extraction:** Compute scalar value estimates $v(s_t)$ as the mean over quantiles using Eq. (15).
9:     **Standard GAE for actor:** Compute scalar advantages $A_t$ using Eq. (16).
10:     **Actor update:** Update policy parameters $\psi$ using the standard PPO clipped surrogate objective with advantages $A_t$.
11: **end for**
**Ensure:** Fine-tuned policy $\pi_\psi$ and distributional critic $q_\theta$.

---

## D. Critic Head Architectures

In this appendix, we summarize the design space of critic head architectures used in this work and related baselines. All heads take as input the critic backbone hidden representation $h_s \in \mathbb{R}^d$ for state $s$ and differ only in how the value function or value distribution is parameterized.

### D.1. Center–Delta Distributional Value Head (Ours)

To ensure that the predicted state-value quantiles $\{q_\theta^{(1)}(s) \le q_\theta^{(2)}(s) \le \cdots \le q_\theta^{(N)}(s)\}$ form a valid (non-crossing) quantile function, we adopt a *center–delta cumulative* parameterization. The head predicts (i) a central reference quantile and (ii) a

sequence of non-negative deltas whose cumulative sums reconstruct the remaining quantiles. This enforces monotonicity *by construction*, without requiring sorting or auxiliary constraints during training.

**Architecture.** Given the critic backbone hidden state $h_s \in \mathbb{R}^d$ for state $s$ (we omit the time index $t$ for simplicity), the value head consists of two bias-free linear projections:

$$c_s = W_c h_s \in \mathbb{R}, \tag{39}$$

$$\Delta_s = W_\Delta h_s \in \mathbb{R}^{N-1}, \tag{40}$$

where $N$ is the number of quantiles. We assume $N$ is odd and define the center index under the same 1-indexed convention as $\{q_\theta^{(i)}(s)\}_{i=1}^N$:

$$m = \frac{N+1}{2}, \qquad L = m - 1, \qquad R = N - m. \tag{41}$$

**Non-negative deltas and monotonicity.** We map the raw delta outputs to non-negative increments via

$$\tilde{\Delta}_s = \text{softplus}(\Delta_s),$$

and (optionally) bound the center output if returns are in $[0, 1]$:

$$\tilde{c}_s = \begin{cases} \sigma(c_s), & \text{if returns are normalized to } [0,1], \\ c_s, & \text{otherwise.} \end{cases}$$

We then split $\tilde{\Delta}_s$ into left and right segments, $\tilde{\Delta}_s^{(\text{L})} \in \mathbb{R}^L$ and $\tilde{\Delta}_s^{(\text{R})} \in \mathbb{R}^R$, and reconstruct the full ordered quantiles by cumulative sums around the center:

$$q_\theta^{(m)}(s) = \tilde{c}_s, \tag{42}$$

$$q_\theta^{(m-r)}(s) = \tilde{c}_s - \sum_{k=1}^{r} \tilde{\Delta}_{s,k}^{(\text{L})}, \qquad r = 1, \ldots, L, \tag{43}$$

$$q_\theta^{(m+r)}(s) = \tilde{c}_s + \sum_{k=1}^{r} \tilde{\Delta}_{s,k}^{(\text{R})}, \qquad r = 1, \ldots, R. \tag{44}$$

Since each increment is non-negative, the resulting quantiles satisfy $q_\theta^{(1)}(s) \leq \cdots \leq q_\theta^{(N)}(s)$ for every state $s$.

### D.2. Standard Scalar Value Head

The standard PPO critic predicts a single scalar value for each state via a linear projection:

$$V(s) = \mathbf{w}_v^\top h_s, \tag{45}$$

where $\mathbf{w}_v \in \mathbb{R}^d$. This head directly regresses the expected return using a mean-squared error objective. While computationally efficient, this formulation collapses the entire return distribution into a single expectation and discards higher-order uncertainty information.

### D.3. Multi-Head Scalar Value Aggregation

An extension of the scalar critic employs multiple independent value heads to increase representational capacity. Specifically, we predict $K$ scalar values $\{v_k(s)\}_{k=1}^K$ via

$$v_k(s) = \mathbf{w}_k^\top h_s, \qquad k = 1, \ldots, K, \tag{46}$$

and combine them through a learned convex combination:

$$V(s) = \sum_{k=1}^{K} \tilde{w}_k \, v_k(s), \qquad \tilde{w}_k = \frac{\exp(w_k)}{\sum_{j=1}^{K} \exp(w_j)}. \tag{47}$$

This architecture can be interpreted as a shallow ensemble over scalar critics. However, since all heads are ultimately collapsed into a single scalar target, it does not explicitly model return distributions.

### D.4. Non-Crossing Quantile Head

Following quantile-based distributional RL, one may parameterize the return distribution using a structured quantile function that enforces monotonicity by construction. This design is closely related to *non-crossing quantile regression* and has been shown to improve stability and representation quality in distributional value learning. In particular, our quantile head is inspired by the non-crossing quantile parameterization proposed by Zhou et al. (2020), and is conceptually aligned with QR-DQN-style value representations.

**Architecture.**  Given the critic backbone hidden representation $h_s \in \mathbb{R}^d$, the head predicts three components through independent linear projections: (i) a set of normalized quantile increments, (ii) a positive scale parameter, and (iii) a location parameter. Concretely,

$$\phi(s) = \text{Softmax}(W_\phi h_s) \in \mathbb{R}^N, \tag{48}$$
$$\alpha(s) = \text{softplus}(W_\alpha h_s) \in \mathbb{R}_+, \tag{49}$$
$$\beta(s) = W_\beta h_s \in \mathbb{R}, \tag{50}$$

where $W_\phi \in \mathbb{R}^{N \times d}$ and $W_\alpha, W_\beta \in \mathbb{R}^{1 \times d}$. The softmax ensures $\phi_i(s) > 0$ and $\sum_{i=1}^N \phi_i(s) = 1$, while the softplus enforces $\alpha(s) > 0$.

**Quantile construction.**  The predicted return distribution is defined via the cumulative quantile function

$$q_\theta^{(i)}(s) = \beta(s) + \alpha(s)\,\psi\left(\sum_{j=1}^i \phi_j(s)\right), \qquad i = 1, \ldots, N, \tag{51}$$

where $\psi(\cdot)$ is a fixed monotone function, such as the identity or logit mapping. Because the cumulative sums satisfy $0 < \sum_{j=1}^1 \phi_j(s) < \cdots < \sum_{j=1}^N \phi_j(s) = 1$ and $\alpha(s) > 0$, the resulting quantiles obey

$$q_\theta^{(1)}(s) < q_\theta^{(2)}(s) < \cdots < q_\theta^{(N)}(s)$$

for every state $s$.

### D.5. Sort-Based Quantile Head

The sort-based quantile head enforces a valid quantile function by predicting unconstrained values and applying an explicit sorting operation. Compared to structured parameterizations, this design minimizes architectural assumptions at the cost of less smooth optimization.

**Architecture.**  Given the critic backbone hidden state $h_s \in \mathbb{R}^d$, the head consists of a single linear projection that outputs $N$ unconstrained scalar values:

$$\tilde{\mathbf{q}}(s) = W_q h_s \in \mathbb{R}^N, \tag{52}$$

where $W_q \in \mathbb{R}^{N \times d}$.

**Quantile construction.**  To enforce monotonicity, the final quantile estimates are obtained by sorting the predicted values:

$$\mathbf{q}_\theta(s) = \text{Sort}(\tilde{\mathbf{q}}(s)), \tag{53}$$

which guarantees

$$q_\theta^{(1)}(s) \leq q_\theta^{(2)}(s) \leq \cdots \leq q_\theta^{(N)}(s).$$

Modern autodifferentiation frameworks support backpropagation through the sort operator via permutation-based Jacobians.

## E. Value Prediction Accuracy via Prefix-Based Monte Carlo Evaluation

**Goal.**  We design a controlled evaluation to compare the *accuracy and stability* of value predictions produced by the scalar PPO critic and the distributional critic in DisPPO. The goal is to assess whether DisPPO yields value estimates that are closer to Monte Carlo return estimates when evaluated on identical intermediate reasoning states at different stages of the generation process.

**Protocol.**  We conduct the evaluation on the AIME25 test set, consisting of 30 problems. For each problem, we sample multiple rollouts from the trained policy using a fixed decoding configuration.

From each rollout, we extract prefix states using two complementary schemes. First, we consider an absolute prefix position at a fixed token index ($t = 500$), which provides a length-controlled view of intermediate reasoning states. Second, we extract prefix states at three relative positions corresponding to 25%, 50%, and 75% of the generated sequence length, representing early-, mid-, and late-stage reasoning contexts.

For each prefix state $s$, we estimate a Monte Carlo value target by resampling from the same prefix. Specifically, we generate $N_{\text{MC}}$ independent continuations conditioned on $s$, compute the terminal reward for each continuation, and take their empirical mean:

$$V^{\text{MC}}(s) \;=\; \frac{1}{N_{\text{MC}}} \sum_{j=1}^{N_{\text{MC}}} R^{(j)}(s), \tag{54}$$

where $R^{(j)}(s)$ denotes the terminal reward of the $j$-th continuation.

In parallel, we compute the value predicted by each critic at the same prefix state. For PPO, this corresponds to the scalar output $V_{\text{PPO}}(s)$. For DisPPO, we obtain a scalar prediction by taking the mean of the predicted quantile distribution:

$$V_{\text{DisPPO}}(s) \;=\; \frac{1}{N} \sum_{i=1}^{N} q_{\theta}^{(i)}(s). \tag{55}$$

**Metric.**  We measure the discrepancy between the predicted value and the Monte Carlo estimate using the mean absolute error (MAE), aggregated over all prefix states at each prefix position:

$$\text{MAE} \;=\; \mathbb{E}_s \big[ \big| V_{\text{pred}}(s) - V^{\text{MC}}(s) \big| \big] . \tag{56}$$

Lower MAE indicates that the critic provides value estimates that more closely match empirical returns at intermediate reasoning states.

**Discussion.**  Across both absolute and relative prefix positions, the distributional critic in DisPPO consistently produces value estimates that are closer to Monte Carlo returns. The improvement is particularly pronounced at the fixed 500-token prefix, suggesting enhanced stability under length-controlled intermediate states. Across relative prefix positions, DisPPO yields lower average error and shows the largest gains at later reasoning stages, where scalar value regression is most prone to error. Together with the binned analysis in Table 7, these results provide direct evidence that modeling the full return distribution leads to more accurate and stable value estimation in long-horizon reasoning.

# F. Detailed Training Configurations

## F.1. Training Data

**Mathematical Reasoning**   For both our train dataset and test dataset, we use the following system prompt:

> **System Prompt**
>
> Please reason step by step, and put your final answer within \boxed{}.

**SQL Generation**   For both our train dataset and test dataset, we do not explicitly use any system prompt. We add the following contents at the beginning of the user prompt:

> **Prompt**
>
> Task Overview:
> You are a data science expert. Below, you are provided with a database schema and a natural language question. Your task is to understand the schema and generate a valid SQL query to answer the question.

**Multi-turn Tool-Augmented Reasoning**   For multi-turn tool-augmented reasoning, we follow the ReTool-style chat template and use the built-in tool instruction prompt. Specifically, we prepend the following system prompt to each conversation in both training and evaluation:

> **System Prompt**
>
> You are Qwen, created by Alibaba Cloud. You are a helpful assistant.
> ```
> # Tools
> ```
> You may call one or more functions to assist with the user query.
> You are provided with function signatures within `<tools></tools>` XML tags:
> ```
> <tools>
> {"type": "function", "function": {"name": "code_interpreter",
> "description": "A tool for executing code.", "parameters": {"type":
> "object", "properties": {"code": {"type": "string", "description": "The
> code to execute."}}, "required": ["code"]}}
> </tools>
> ```
> For each function call, return a JSON object with function name and arguments within `<tool_call></tool_call>` XML tags:
> ```
> <tool_call>
> {"name": <function-name>, "arguments": <args-json-object>}
> </tool_call>
> ```

## F.2. RL Training Configuration

**Mathematical Reasoning**   We use the hyperparameters in Table 8 for RL training.

We use an outcome-based reward function that assigns +1 for correct final answers and 0 otherwise.

**SQL Generation**   We use the hyperparameters in Table 10 for RL training on SQL generation tasks.

The outcome-based reward function is dense: $\text{final\_score} = \text{answer\_score} + \text{format\_score}$, where:

$$\text{answer\_score} = \begin{cases} 1.0, & \text{if } \text{Result}(S) = \text{Result}(G) \\ \min\left(\frac{\text{count}^2}{|\text{gold\_dict}| \times |\text{result\_dict}|}, 1.0\right) \times 0.8 & \text{if } \text{Result}(S) \neq \text{Result}(G) \end{cases} \tag{57}$$

Above, $S$ is the generated solution string (predicted SQL query), and $G$ is the ground truth query. $\text{Result}(Q)$ is the set of execution results returned by the database when executing the SQL query $Q$.

*Table 8.* RL training base configurations.

| Hyperparameter | Value |
| --- | --- |
| Optimizer | AdamW |
| Policy learning rate | 1e-6 |
| Training batch size | 512 |
| Samples per prompt | 16 |
| Mini-batch size | 32 |
| Max prompt length | 1024 |
| Max response length | 3072 |
| Rollout temperature | 1.0 |

*Table 9.* Our RL training configurations on Multi-turn Tool-Augmented Reasoning.

| Hyperparameter | Value |
| --- | --- |
| Optimizer | AdamW |
| Policy learning rate | 1e-6 |
| Training batch size | 128 |
| Samples per prompt | 16 |
| Mini-batch size | 32 |
| Max prompt length | 2048 |
| Max response length | 16384 |
| Max turns 8 | |
| Rollout temperature | 1.0 |

*Table 10.* Our RL training configurations on SQL generation tasks.

| Hyperparameter | Value |
| --- | --- |
| Optimizer | AdamW |
| Policy learning rate | 1e-6 |
| Training batch size | 128 |
| Samples per prompt | 8 |
| Mini-batch size | 64 |
| Max prompt length | 8192 |
| Max response length | 4096 |
| Rollout temperature | 1.0 |

*Table 11.* PPO critic training configurations.

| Hyperparameter | Value |
|---|---|
| Optimizer | AdamW |
| Critic learning rate | $2 \times 10^{-6}$ |
| Discount factor $\gamma$ | 1 |
| GAE parameter $\lambda$ | 1 |
| Gradient clipping | 1.0 |
| PPO clip range | 0.5 |
| Number of critic updates per epoch | 1 |
| Critic warmup steps | 30 |

*Table 12.* DisPPO-specific critic configurations.

| Hyperparameter | Value |
|---|---|
| Number of quantiles $N$ | 51 |
| Quantile Huber loss threshold $\kappa$ | 1 |
| Quantile parameterization | Center-delta |

**Multi-turn Tool-Augmented Reasoning**   For multi-turn tool-augmented reasoning, we define an outcome-based reward with an auxiliary tool-usage term. Let $s$ denote the generated solution and $y$ the ground-truth answer. We first compute a base correctness score using strict boxed-answer verification:

$$r_{\text{ans}}(s, y) \in \{-1, 1\}, \tag{58}$$

where $r_{\text{ans}}(s, y) = 1$ if the final boxed answer is correct and 0 otherwise.

To account for tool usage, we define a lightweight turn-based bonus. Let $T$ denote the total number of interaction turns in the trajectory. The tool-related adjustment is given by

$$r_{\text{tool}}(T) = \frac{T - 2}{2} \cdot \alpha, \tag{59}$$

where $\alpha = 0.1$ is a fixed scaling constant and the first two turns receive no bonus.

The final reward is computed as

$$r(s, y, T) = \begin{cases} r_{\text{ans}}(s, y) + \mathbb{I}[r_{\text{tool}}(T) > 0], & \text{if } r_{\text{ans}}(s, y) = 1, \\ \max(0, \ r_{\text{ans}}(s, y) + r_{\text{tool}}(T)), & \text{otherwise}, \end{cases} \tag{60}$$

where the auxiliary term is bounded so that incorrect final answers do not receive positive reward.

**Entropy-based advantage shaping configurations.**   For all entropy-based advantage shaping baselines, we adopt the same formulation as in prior work (Cheng et al., 2025). We fix the temperature parameter $\kappa = 2$ across all experiments. The entropy coefficient $\alpha$ is set to $0.4$ for GRPO and $0.1$ for PPO, following the recommended settings in the original implementations. These hyperparameters are kept fixed throughout training and are not tuned per task or per benchmark, ensuring a fair and consistent comparison.

**Critic configuration details**   We provide here the complete hyperparameter settings used in our experiments. Our method extends the PPO foundation with additional distributional components.

**DisPPO critic.**   DisPPO extends the PPO critic with a distributional value head while keeping the same optimization and update schedule. Table 12 summarizes the additional hyperparameters specific to the distributional critic.

*Table 13.* Avg@*k* results on mathematical reasoning benchmarks over three random seeds. We report mean±std. DisPPO consistently outperforms both PPO and VAPO on average.

| Method | AIME24 | College | MATH500 | Minerva | OlympiadBench | AIME25 | Avg. |
|---|---|---|---|---|---|---|---|
| PPO | $32.5 \pm 1.1$ | $42.0 \pm 0.4$ | $81.1 \pm 0.1$ | $44.5 \pm 0.2$ | $43.0 \pm 0.3$ | $16.6 \pm 0.4$ | $43.3 \pm 0.2$ |
| VAPO | $33.0 \pm 1.9$ | $41.9 \pm 0.3$ | $81.3 \pm 0.2$ | $45.0 \pm 0.2$ | $42.7 \pm 0.4$ | $18.0 \pm 0.4$ | $43.6 \pm 0.3$ |
| DisPPO (ours) | $\mathbf{36.4 \pm 1.4}$ | $\mathbf{42.8 \pm 0.3}$ | $\mathbf{82.2 \pm 0.1}$ | $\mathbf{46.7 \pm 0.2}$ | $\mathbf{44.1 \pm 0.3}$ | $\mathbf{19.3 \pm 0.5}$ | $\mathbf{45.3 \pm 0.3}$ |

## G. Inference Configurations

**Mathematical Reasoning**   We use a rollout temperature of 0.6, top-$p$ sampling with $p = 0.95$, and a maximum response length of 4096 tokens. We adopt $k = 256$ for the small but challenging AIME2024/2025 datasets (30 problems each), $k = 16$ for Minerva, MATH500, and OlympiadBench, and $k = 8$ for College Math, balancing computational cost and difficulty.

**SQL Generation**   We use a top-$p$ sampling with $p = 0.95$, and a maximum response length of 4096 tokens. We use a rollout temperature of 0.0 for greedy sampling, and a rollout temperature of 1.0 to evaluate pass@$k$.

**Multi-turn Tool-Augmented Reasoning.**   We use a rollout temperature of 0.6, top-$p$ sampling with $p = 0.95$, and a maximum response length of 16384 tokens.

## H. Additional Ablation Studies

In this appendix, we provide additional ablation studies that complement the main experimental results. These analyses focus on (i) robustness against a strong VAPO baseline over repeated runs, (ii) the design choices of the distributional critic, and (iii) the sensitivity of DisPPO to the number of quantiles in distributional value modeling.

### H.1. VAPO Comparison over Multiple Random Seeds

To address whether DisPPO's gains come from comparing against weak PPO baselines, we additionally compare against VAPO (Yue et al., 2025), a recent value-regularized PPO variant for reasoning tasks. We repeat PPO, VAPO, and DisPPO with three different random seeds under the same mathematical reasoning setup and report avg@$k$ mean±std in Table 13. Our PPO baseline already follows a strong large-scale RL-for-reasoning recipe, incorporating optimization techniques also used by VAPO, including Clip-Higher, token-level loss, critic pretraining, and group sampling. The entropy-regularized PPO variant in the main paper is applied on top of this strengthened PPO baseline.

These results show that DisPPO's improvements are stable across random seeds and remain positive relative to VAPO, supporting that the gains arise from improved value learning rather than from comparison against a weak PPO baseline.

### H.2. Ablation on Critic Design for Mathematical Reasoning

We conduct a systematic ablation on the critic design for mathematical reasoning tasks, evaluating different combinations of value head architectures and training objectives on AIME24 and AIME25. All models are evaluated using pass@*256* and avg@*256*.

**Effect of the Training Objective.**   We first isolate the effect of the critic training objective while keeping the architecture fixed. Using the same center–delta value head, we replace quantile regression with a mean-squared error (MSE) loss applied to the averaged prediction. As shown in Table 14, this variant consistently underperforms the full DisPPO model on both AIME24 and AIME25 in terms of avg@*256* and pass@*256*. This result indicates that the observed gains are not due to the head structure alone, but critically depend on learning the full return distribution via quantile regression rather than a scalar objective.

**Effect of the Critic Architecture.**   Next, we compare different critic head architectures under the same quantile regression objective. Specifically, Table 14 reports results for the proposed center–delta head, a non-crossing quantile head, and a sort-based quantile head, all of which enforce monotonicity by construction. While both alternative quantile heads outperform

*Table 14.* Ablation on critic architectures and training objectives for mathematical reasoning on AIME24 and AIME25. All results are reported using pass@*256* and avg@*256*.

| Critic Head | Loss | AIME24 | | AIME25 | |
|---|---|---|---|---|---|
| | | pass@*256* | avg@*256* | pass@*256* | avg@*256* |
| Scalar Head | MSE | 73.7 | 31.3 | 60.0 | 16.4 |
| Multi-Head Scalar | MSE | 63.3 | 32.9 | 60.0 | 18.4 |
| Non-Crossing Quantile | Quantile | 63.3 | 32.9 | **70.0** | 17.4 |
| Sort-Based Quantile | Quantile | 80.0 | **35.1** | 60.0 | 17.5 |
| Center–Delta | MSE | 76.7 | 33.2 | 53.3 | 17.1 |
| Center–Delta (Ours) | Quantile | **86.7** | 33.7 | 63.3 | **19.2** |

*Table 15.* Ablation on the number of quantiles $N$ for DisPPO on SQL generation. Best results in **bold**, second best underlined.

| $N$ Quantiles | **Bird** (in domain) | | | **Spider** (out of domain) | | | Avg |
|---|---|---|---|---|---|---|---|
| | Greedy | Pass@8 | Pass@16 | Greedy | Pass@8 | Pass@16 | |
| 17 | 61.4 | 74.2 | 77.2 | 77.9 | 85.4 | 86.7 | 77.1 |
| 35 | 62.7 | 74.4 | 77.4 | 78.1 | 86.1 | 86.9 | 77.9 |
| 51 | 63.5 | 75.0 | **77.4** | 81.2 | 87.5 | **88.9** | 78.9 |
| 101 | **63.6** | **75.2** | **77.4** | **81.3** | **87.6** | **88.9** | **79.0** |

scalar critics, they consistently lag behind the center–delta design across AIME24 and AIME25. This suggests that although enforcing quantile monotonicity is necessary for valid distributional value estimation, the specific parameterization of the quantile function plays an important role in long-horizon reasoning tasks.

**Scalar Critics with Increased Capacity.** Finally, we evaluate two scalar baselines reported in Table 14: a standard single-head value function and a multi-head scalar aggregation trained with MSE. Despite the increased representational capacity of the multi-head variant, both scalar critics remain significantly worse than all distributional critics in avg@*256* on AIME24 and AIME25. This confirms that the performance gains of DisPPO cannot be attributed to model capacity alone, but instead arise from explicitly modeling the return distribution.

### H.3. Ablation on the Number of Quantiles

We study the sensitivity of DisPPO to the number of quantiles used to parameterize the value distribution. Following prior work in distributional reinforcement learning, the quantile count controls the resolution of the learned return distribution. We evaluate four representative settings:

$$N \in \{17, 35, 51, 101\}.$$

All experiments follow the same setup as the main SQL generation evaluation: models are trained on the BIRD training set and evaluated on both BIRD (in-domain) and Spider (out-of-domain). Only the number of quantiles is varied to isolate its effect.

**Findings.** Increasing the number of quantiles consistently improves performance from $N = 17$ to $N = 35$, and further to $N = 51$, with particularly strong gains on the out-of-domain Spider benchmark. Beyond $N = 51$, the improvements begin to saturate, and the gains from additional quantiles become marginal.

**Conclusion.** DisPPO is robust to the choice of quantile count across a wide range of settings. Using $N = 51$ provides a strong performance–efficiency trade-off and is adopted as the default configuration in our experiments.

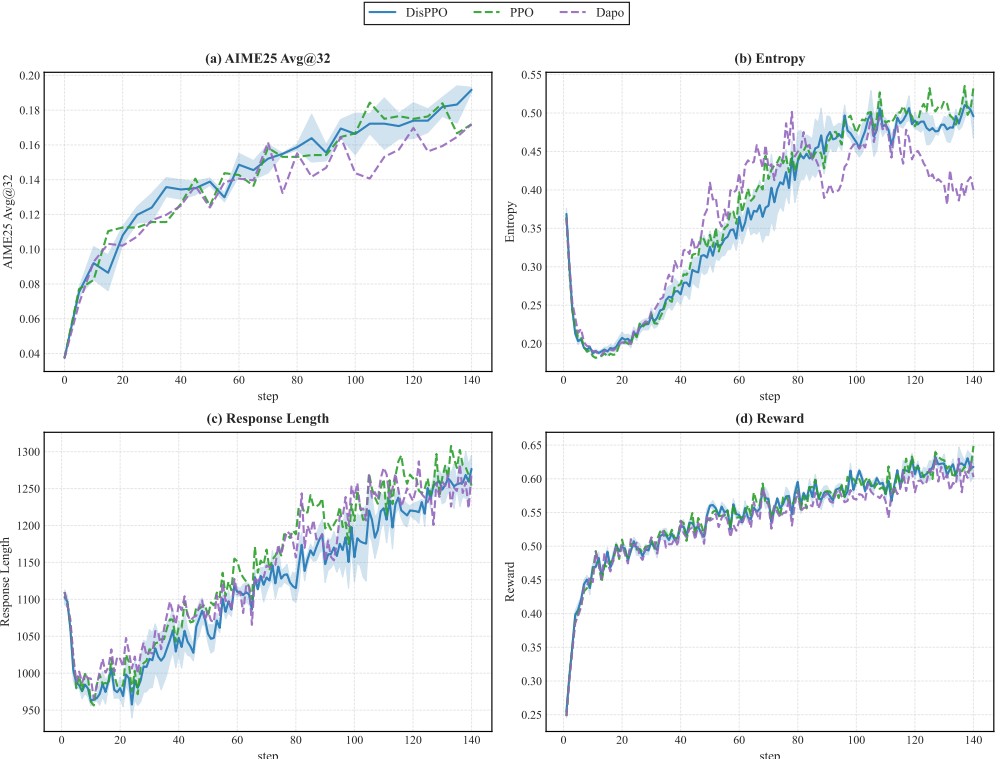

*Figure 2.* Training dynamics on Mathematical Reasoning. We report (a) avg@32 accuracy, (b) policy entropy, (c) response length, and (d) reward as a function of training steps. Results for DisPPO are averaged over three random seeds; shaded regions indicate variance across runs.

# I. Training Dynamics

Figure 2 illustrates the training dynamics of DisPPO in comparison with PPO and DAPO on the Mathematical Reasoning task. For DisPPO, all experiments are conducted with three different random seeds. We report the evolution of (a) avg@32 accuracy, (b) policy entropy, (c) response length, and (d) reward over training steps. All curves are smoothed with a moving average, and shaded regions indicate variance across runs.

**Accuracy (Avg@32).** As shown in Figure 2(a), all methods exhibit rapid performance improvement in the early stage of training, reflecting effective initial policy optimization. DisPPO consistently tracks or slightly outperforms PPO throughout training and achieves the highest final avg@32 score. Compared to DAPO, DisPPO demonstrates more stable improvement with reduced oscillation in later stages, suggesting more reliable credit assignment during long-horizon reasoning.

**Policy Entropy.** Figure 2(b) shows the evolution of policy entropy. All methods experience an initial entropy drop as the policy quickly adapts from the pretrained initialization, followed by a gradual increase as training progresses. DisPPO maintains entropy levels comparable to PPO, indicating that the distributional critic does not induce premature policy collapse. In contrast, DAPO exhibits larger entropy fluctuations in later stages, which aligns with its higher variance in performance.

**Response Length.** Figure 2(c) reports the average response length. All methods initially shorten their outputs before gradually increasing response length as reasoning depth improves. DisPPO produces slightly shorter responses than PPO and DAPO on average, suggesting more concise reasoning without sacrificing accuracy. This indicates that the performance gains of DisPPO are not driven by longer or more verbose generations.

**Reward.** As shown in Figure 2(d), the reward curves for all methods increase steadily and largely overlap. DisPPO achieves comparable or marginally higher rewards while exhibiting smoother trajectories, indicating stable optimization. Notably, the

*Table 16.* Results of cross-domain experiments on MMLU-Pro, GPQA, and ARC. Best results are in **bold** and second best are underlined.

| Model | GPQA | | | MMLU-Pro | ARC | |
|---|---|---|---|---|---|---|
| | mean@16 | pass@8 | pass@16 | pass@1 | mean@16 | pass@16 |
| *Qwen2.5-Math* | 8.98 | 47.5 | 53.6 | 5.8 | 45.1 | 13.3 |
| + GRPO | 24.3 | 59.8 | 61.6 | 28.3 | 75.4 | 91.3 |
| + DAPO | 26.2 | 61.2 | **70.5** | 37.4 | 77.4 | 93.2 |
| + PPO | 25.9 | 60.1 | 65.4 | 33.5 | 76.3 | 92.5 |
| + DisPPO (ours) | **26.9** | **62.1** | 67.4 | **38.3** | **78.1** | **93.3** |

differences in final reasoning performance (Figure 2(a)) are not solely explained by reward magnitude, highlighting the role of improved value estimation in shaping learning dynamics.

**Summary.**  Overall, the training curves demonstrate that DisPPO improves reasoning accuracy without introducing instability in entropy, response length, or reward. Across three random seeds, DisPPO exhibits reduced variance and smoother learning trajectories, supporting the claim that distributional value estimation provides a more informative and stable learning signal than scalar critics.

## J. Cross-domain Experiments

To further evaluate the cross-domain effectiveness of our approach, we transfer the RL models trained on the Mathematical Reasoning dataset to downstream tasks such as MMLU-Pro, GPQA, and the ARC Challenge Set (test split) for testing. We convert each problem into a multiple-choice question (MCQ) format, and the system prompt is as follows. For GPQA and ARC, we sample up to 16 times, whereas for MMLU-Pro, we sample only once due to its large scale. The results are shown in Table 16.

> **System Prompt**
>
> What of the following is the right choice? Please reason step by step, and put your final answer within \boxed{}. The final answer must be a capital letter like A, B, C, or D.

These findings suggest that our DisPPO plays an important role in improving out-of-domain robustness, even when the underlying training data is highly domain-specialized.

## K. Comparison with Parametric Distributional Critics

This section provides an empirical clarification of the design choice to adopt a non-parametric distributional critic in DisPPO. While the main paper motivates this choice based on the characteristics of reasoning rewards, the discussion necessarily abstracts away from alternative parametric formulations. Here, we explicitly compare DisPPO with a parametric Gaussian Mixture Model (GMM)-based distributional critic, following the GMAC-style approach, under the same large language model reasoning setting.

The goal of this analysis is not to question the expressive power of parametric distributional models in general, but to examine their practical behavior when applied to on-policy value estimation with sparse and non-stationary returns. By isolating the critic architecture and holding the actor, data, and training protocol fixed, this comparison aims to assess whether non-parametric quantile regression offers advantages in stability and value estimation quality for LLM reasoning tasks.

**Motivation.**  The main text motivates the use of a non-parametric distributional critic by arguing that parametric assumptions may be ill-suited for reasoning tasks. While parametric models such as Gaussian Mixture Models (GMMs) are expressive and capable of representing multi-modal distributions in principle, their suitability in the context of large language model (LLM) reasoning remains an open empirical question.

In LLM reasoning tasks, return distributions exhibit several distinctive characteristics. Rewards are typically sparse and

*Table 17.* GMAC critic configuration used in mathematical reasoning experiments.

| Parameter | Value | Description |
|---|---|---|
| Discount factor $\gamma$ | 0.99 | Reward discounting coefficient |
| TD parameter $\lambda$ | 0.95 | Trace decay parameter in SR($\lambda$) |
| Number of SR($\lambda$) samples | 64 | Samples used to construct the distributional target |
| Number of GMM components ($K$) | 8 | Mixture components in the value distribution |
| Minimum variance | $1 \times 10^{-4}$ | Variance floor for numerical stability |

low-entropy, often approximating binary correctness signals. Moreover, value estimation is performed in an on-policy, bootstrapped setting, where critic targets are highly non-stationary and depend strongly on intermediate policy behavior. Under these conditions, parametric likelihood-based critics may suffer from instability, sensitivity to initialization, or mode collapse, despite their theoretical expressiveness.

To clarify whether the advantage of DisPPO arises from the use of non-parametric quantile regression rather than from other design choices, we conduct a direct comparison with a parametric GMM-based distributional critic following the GMAC-style formulation. This experiment is designed to evaluate the practical stability and value estimation quality of parametric versus non-parametric distributional critics under the same LLM reasoning setup.

**GMAC Critic Architecture and Training Objective.** Following GMAC, we parameterize the state value distribution $Z_\theta(s)$ as a $K$-component Gaussian mixture model: $Z_\theta(s) = \sum_{k=1}^{K} w_k(s) \mathcal{N}(\mu_k(s), \sigma_k^2(s))$, where the mixture weights $w_k(s)$ are normalized via softmax and variances are enforced to be positive. The critic backbone is identical to PPO, with only the value head replaced by the GMM parameterization.

GMAC constructs multi-step distributional Bellman targets using the Sample-Replacement TD($\lambda$) (SR($\lambda$)) algorithm, which produces a distributional analogue of the $\lambda$-return by mixing $n$-step return distributions. The resulting target distribution $Z_t^{(\lambda)}$ is represented in the same GMM form.

The critic is trained by minimizing the energy (Cramér) distance between the predicted distribution $Z_\theta(s_t)$ and the target distribution $Z_t^{(\lambda)}$. For Gaussian mixtures, this distance admits a closed-form expression, enabling efficient and unbiased gradient-based optimization. During policy optimization, the actor uses the expectation of the learned value distribution, $\mathbb{E}[Z_\theta(s)]$, to compute advantages, yielding a standard actor–critic update.

### K.1. Experimental Setup: GMAC vs. DisPPO on Mathematical Reasoning

**Task and evaluation.** We compare GMAC and DisPPO on mathematical reasoning, following the same benchmarks, decoding configuration, and evaluation metrics (avg@$k$) as in the main mathematical reasoning experiments.

**Common training configuration.** To ensure a fair comparison, GMAC and DisPPO share the same backbone model, training data, prompt formatting, reward function, rollout length/truncation rules, batch size, optimizer, learning-rate schedule, and PPO-style actor update (including clipping, $\gamma$, and $\lambda$). Unless otherwise stated, all hyperparameters are identical to those used in the mathematical reasoning setting in the main paper; we only change the critic parameterization and the corresponding critic learning objective.

**DisPPO configuration.** DisPPO uses the distributional critic described in the main paper, where the value distribution is represented by $N$ ordered quantiles and trained via quantile Huber regression with Distributional GAE (D-GAE). All DisPPO hyperparameters (e.g., number of quantiles $N$) are exactly the same as in the main mathematical reasoning experiments.

**GMAC configuration.** For GMAC, we replace the DisPPO quantile-based critic head with a Gaussian mixture model (GMM) value head consisting of $K$ components. The critic backbone is kept identical to the actor backbone (as in our PPO/DisPPO implementations), and only the value head and critic loss are changed. We train the GMAC critic using the Sample-Replacement TD($\lambda$) (SR($\lambda$)) target construction and the distributional regression objective as in the GMAC formulation. Following the original GMAC setup, we use a discount factor $\gamma = 0.99$ and a TD parameter $\lambda = 0.95$ for GMAC. In our implementation, we use configuration in the Table 17

*Table 18.* Comparison between GMAC and DisPPO on mathematical reasoning. All results are reported using pass@*256* and avg@*256* on AIME24 and AIME25.

| Method | AIME24 | | AIME25 | |
|---|---|---|---|---|
| | pass@*256* | avg@*256* | pass@*256* | avg@*256* |
| GMAC | 63.3 | 32.3 | 53.3 | 16.2 |
| DisPPO (Ours) | 86.7 | 33.7 | 63.3 | 19.2 |

**Fairness and controlled differences.** Both methods optimize the same expected-return objective and use the same actor update rule; the only intended difference is the *critic modeling choice* and its associated target/loss: DisPPO models $Z(s)$ via non-parametric quantiles and trains with quantile regression under D-GAE, whereas GMAC models $Z(s)$ via a parametric GMM and trains using $SR(\lambda)$-based distributional targets. This isolates the effect of distribution parameterization (quantiles vs. GMM) under an otherwise matched PPO-style training pipeline.

**Results and discussion.** Table 18 reports the comparison between GMAC and DisPPO on AIME24 and AIME25. Under a fully matched PPO-style training pipeline, DisPPO consistently outperforms GMAC across both benchmarks and evaluation metrics. On AIME24, DisPPO achieves substantial gains in pass@*256* (86.7 vs. 63.3) and improves avg@*256* from 32.3 to 33.7. Similar trends are observed on AIME25, where DisPPO improves pass@*256* from 53.3 to 63.3 and avg@*256* from 16.2 to 19.2. These results indicate that, under identical training conditions,

## L. Compute, Training Cost

This appendix analyzes the computational cost of DisPPO and its baselines. To ensure implementation-agnostic comparison, we report **theoretical FLOPs** rather than wall-clock time. All numbers correspond to **forward + backward FLOPs per sequence**.

**FLOPs measurement protocol.** We measure forward-pass FLOPs using the `DeepSpeed` FLOPs profiler. Following standard practice (Li et al., 2020), we approximate the backward pass as costing $2\times$ the forward FLOPs. Therefore, the reported "forward + backward" FLOPs equal $3\times$ the measured forward FLOPs.

**Training vs. inference cost.** At inference time, only the actor model is used. Consequently, **inference FLOPs are identical across all methods** and correspond to the **GRPO (Actor)** column in the tables below. Both PPO and DisPPO incur additional FLOPs *only during training* due to the inclusion of a value model.

**Actor–critic decomposition.** For clarity, we distinguish between two sources of computational overhead: (i) the *total training FLOPs* of the full Actor + Critic pipeline, and (ii) the *critic-only FLOPs*, comparing a distributional critic to a standard scalar value head. The former determines the overall training cost, while the latter isolates the overhead introduced specifically by distributional value modeling.

**Llama-3.1-8B-Instruct.** Table 19 reports the total training FLOPs (Actor + Critic) per sequence at different context lengths. Adding a standard scalar value model (PPO) increases training FLOPs by approximately $1.93\times$ relative to GRPO, reflecting the cost of the critic. Replacing the scalar critic with a distributional critic (DisPPO) results in a total training FLOPs increase of exactly $2.00\times$ relative to GRPO, corresponding to an additional overhead of only $\approx 3.6\%$ over PPO. Importantly, inference cost remains unchanged.

Table 20 further isolates the critic computation. It compares the FLOPs of a distributional value head against a standard scalar value head *within the critic alone*. The distributional critic increases the **critic-only training FLOPs** by approximately $7.53\%$ across all sequence lengths. Because the actor dominates overall computation, this translates to a much smaller increase in total training FLOPs, as shown in Table 19.

**Qwen2.5-Math-7B.** Table 21 presents the corresponding results for Qwen2.5-Math-7B. As with Llama-3.1-8B, PPO increases training FLOPs by approximately $1.92\times$ over GRPO due to the scalar value model. DisPPO increases total

training FLOPs by exactly $2.00\times$ relative to GRPO, amounting to an additional overhead of approximately $4.0\%$ over PPO, while leaving inference cost unchanged.

Table 22 reports the critic-only FLOPs comparison. The distributional value head increases the **critic-only training FLOPs** by approximately $8.36\%$ relative to a scalar critic. Again, because the critic constitutes a minority of the overall computation, this overhead corresponds to only a $\approx 4\%$ increase in total training FLOPs, as summarized in Table 21.

**Summary.** Across both backbones, DisPPO introduces a modest computational overhead during training—approximately $3.6$–$4.0\%$ in total FLOPs compared to PPO—while keeping inference cost identical. This demonstrates that modeling return distributions via a quantile-based critic provides improved learning signal at minimal additional computational cost, making DisPPO practical for large-scale LLM fine-tuning.

*Table 19.* Llama3.1-8B-Instruct: total FLOPs (forward + backward) per sequence at different context lengths. GRPO uses only an actor; PPO adds a standard value model; DisPPO adds a distributional value model. Inference cost corresponds to the **GRPO (Actor)** column.

| Seq Len | GRPO (Actor) | PPO (A+V) | DisPPO (A+DV) | PPO/GRPO | DisPPO/GRPO | DisPPO/PPO |
|---|---|---|---|---|---|---|
| 128 | 5.76T | 11.12T | 11.53T | $1.9300\times$ | $2.0000\times$ | $1.036285\times$ |
| 256 | 11.53T | 22.25T | 23.06T | $1.9300\times$ | $2.0000\times$ | $1.036285\times$ |
| 512 | 23.06T | 44.50T | 46.11T | $1.9300\times$ | $2.0000\times$ | $1.036285\times$ |
| 1024 | 46.11T | 88.99T | 92.22T | $1.9300\times$ | $2.0000\times$ | $1.036285\times$ |
| 2048 | 92.22T | 177.98T | 184.44T | $1.9300\times$ | $2.0000\times$ | $1.036285\times$ |

*Table 20.* Llama3.1-8B-Instruct: standard value vs. distributional value FLOPs (forward + backward).

| Seq Len | Std Fwd | Std Bwd | Dist Fwd | Dist Bwd | Total Ratio |
|---|---|---|---|---|---|
| 128 | 1.79T | 3.57T | 1.92T | 3.84T | $1.075302\times$ |
| 256 | 3.57T | 7.15T | 3.84T | 7.69T | $1.075302\times$ |
| 512 | 7.15T | 14.29T | 7.69T | 15.37T | $1.075302\times$ |
| 1024 | 14.29T | 28.59T | 15.37T | 30.74T | $1.075302\times$ |
| 2048 | 28.59T | 57.18T | 30.74T | 61.48T | $1.075302\times$ |

*Table 21.* Qwen2.5-Math-7B: total FLOPs (forward + backward) per sequence at different context lengths.

| Seq Len | GRPO (Actor) | PPO (A+V) | DisPPO (A+DV) | PPO/GRPO | DisPPO/GRPO | DisPPO/PPO |
|---|---|---|---|---|---|---|
| 128 | 5.43T | 10.44T | 10.86T | $1.9229\times$ | $2.0000\times$ | $1.040100\times$ |
| 256 | 10.86T | 20.88T | 21.72T | $1.9229\times$ | $2.0000\times$ | $1.040100\times$ |
| 512 | 21.72T | 41.77T | 43.44T | $1.9229\times$ | $2.0000\times$ | $1.040100\times$ |
| 1024 | 43.44T | 83.53T | 86.88T | $1.9229\times$ | $2.0000\times$ | $1.040100\times$ |
| 2048 | 86.88T | 167.07T | 173.77T | $1.9229\times$ | $2.0000\times$ | $1.040100\times$ |

## M. Implementation Details and Code Release

We release the reference implementation of DisPPO to facilitate reproducibility. The codebase includes our distributional critic, D-GAE target construction, and all training/evaluation scripts used in the experiments.

**Repository.** Our implementation is available at: https://github.com/ZhijianZhou/Disppo

*Table 22.* Qwen2.5-Math-7B: standard value vs. distributional value FLOPs (forward + backward).

| Seq Len | Std Fwd | Std Bwd | Dist Fwd | Dist Bwd | Total Ratio |
|---------|---------|---------|----------|----------|-------------|
| 128 | 1.67T | 3.34T | 1.81T | 3.62T | 1.083550× |
| 256 | 3.34T | 6.68T | 3.62T | 7.24T | 1.083550× |
| 512 | 6.68T | 13.36T | 7.24T | 14.48T | 1.083550× |
| 1024 | 13.36T | 26.73T | 14.48T | 28.96T | 1.083550× |
| 2048 | 26.73T | 53.46T | 28.96T | 57.92T | 1.083550× |

# N. Case Study

We analyzed the cases on AIME2025 to examine the effect of incorporating our method, and the results are as follows.

---

**Case Study**

**Question:** Find the number of ordered pairs $(x, y)$, where both $x$ and $y$ are integers between $-100$ and $100$, inclusive, such that $12x^2 - xy - 6y^2 = 0$

---

**Response from Qwen-2.5-Math-7B-Base:**
To solve the problem, we need to find the number of ordered pairs $(x, y)$ that satisfy the equation $12x^2 - xy - 6y^2 = 0$, where both $x$ and $y$ are integers between $-100$ and $100$ inclusive. First, we can factor the equation $12x^2 - xy - 6y^2 = 0$. By factoring, we get:

$$12x^2 - xy - 6y^2 = (4x - 3y)(3x + 2y) = 0.$$

This gives us two equations:

$$4x - 3y = 0 \quad \text{or} \quad 3x + 2y = 0.$$

We can solve each equation separately for $y$ in terms of $x$: 1. $4x - 3y = 0 \implies y = \frac{4x}{3}$. 2. $3x + 2y = 0 \implies y = -\frac{3x}{2}$. For $y = \frac{4x}{3}$, $x$ must be a multiple of 3, and for $y = -\frac{3x}{2}$, $x$ must be a multiple of 2. We need to count the number of valid integer pairs $(x, y)$ for each case. Let's write the Python code to count the number of valid pairs $(x, y)$ for both cases.

**Python Code**

```
# Initialize the count of valid pairs count = 0
# Check for the case y = (4x)/3
for x in range(-100, 101):
    if x % 3 == 0:
        y = (4 * x) // 3
    if -100 <= y <= 100:
        count += 1
# Check for the case y = (-3x)/2
for x in range(-100, 101):
    if x % 2 == 0:
        y = (-3 * x) // 2
    if -100 <= y <= 100:
        count += 1
# Since (0, 0) is counted twice, we need to subtract 1
count -= 1
print(count)
```

The number of ordered pairs $(x, y)$ that satisfy the equation $12x^2 - xy - 6y^2 = 0$ where both $x$ and $y$ are integers between $-100$ and $100$ inclusive is $\boxed{106}$.

**Response from PPO:**

Let's solve the given equation $12x^2 - xy - 6y^2 = 0$ for integer solutions $(x, y)$ where both $x$ and $y$ are in the range from $-100$ to $100$. First, we can factorize the equation by treating it as a quadratic in $x$:

$$12x^2 - xy - 6y^2 = 0$$

We can solve for $x$ in terms of $y$ by using the quadratic formula $x = \frac{-b \pm \sqrt{b^2 - 4ac}}{2a}$, where $a = 12$, $b = -y$, and $c = -6y^2$:

$$x = \frac{-(-y) \pm \sqrt{(-y)^2 - 4 \cdot 12 \cdot (-6y^2)}}{2 \cdot 12} = \frac{y \pm \sqrt{y^2 + 288y^2}}{24} = \frac{y \pm \sqrt{289y^2}}{24} = \frac{y \pm 17|y|}{24}$$

This gives us two potential solutions: 1. $x = \frac{y+17|y|}{24}$ 2. $x = \frac{y-17|y|}{24}$ Let's consider these cases separately.
**Case 1**: $x = \frac{y+17|y|}{24}$ If $y \geq 0$, then $|y| = y$, and the equation becomes:

$$x = \frac{y + 17y}{24} = \frac{18y}{24} = \frac{3y}{4}$$

So $x = \frac{3y}{4}$. For $x$ to be an integer, $y$ must be a multiple of 4. Let's denote $y = 4k$ for some integer $k$. Then:

$$x = \frac{3(4k)}{4} = 3k$$

Hence, the solutions in this case are $(x, y) = (3k, 4k)$. We need $x$ and $y$ to be in the range $[-100, 100]$:

$$-100 \leq 3k \leq 100 \implies -\frac{100}{3} \leq k \leq \frac{100}{3}$$

Since $k$ is an integer:

$$-33 \leq k \leq 33$$

So there are $33 - (-33) + 1 = 67$ possible values for $k$.
**Case 2**: $x = \frac{y-17|y|}{24}$ If $y \geq 0$, then $|y| = y$, and the equation becomes:

$$x = \frac{y - 17y}{24} = \frac{-16y}{24} = -\frac{2y}{3}$$

So $x = -\frac{2y}{3}$. For $x$ to be an integer, $y$ must be a multiple of 3. Let's denote $y = 3m$ for some integer $m$. Then:

$$x = -\frac{2(3m)}{3} = -2m$$

Hence, the solutions in this case are $(x, y) = (-2m, 3m)$. We need $x$ and $y$ to be in the range $[-100, 100]$:

$$-100 \leq 3m \leq 100 \implies -\frac{100}{3} \leq m \leq \frac{100}{3}$$

Since $m$ is an integer:

$$-33 \leq m \leq 33$$

So there are $33 - (-33) + 1 = 67$ possible values for $m$. If $y < 0$, then $|y| = -y$, and the equation becomes:

$$x = \frac{y + 17(-y)}{24} = \frac{y - 17y}{24} = \frac{-16y}{24} = -\frac{2y}{3}$$

So $x = -\frac{2y}{3}$. For $x$ to be an integer, $y$ must be a multiple of 3. Let's denote $y = 3n$ for some integer $n$. Then:

$$x = -\frac{2(3n)}{3} = -2n$$

Hence, the solutions in this case are $(x, y) = (-2n, 3n)$. We need $x$ and $y$ to be in the range $[-100, 100]$:

$$-100 \leq 3n \leq 100 \implies -\frac{100}{3} \leq n \leq \frac{100}{3}$$

Since $n$ is an integer:

$$-33 \leq n \leq 33$$

So there are $33 - (-33) + 1 = 67$ possible values for $n$. If $y < 0$, then $|y| = -y$, and the equation becomes:

$$x = \frac{y - 17(-y)}{24} = \frac{y + 17y}{24} = \frac{18y}{24} = \frac{3y}{4}$$

So $x = \frac{3y}{4}$. For $x$ to be an integer, $y$ must be a multiple of 4. Let's denote $y = 4p$ for some integer $p$. Then:

$$x = \frac{3(4p)}{4} = 3p$$

Hence, the solutions in this case are $(x, y) = (3p, 4p)$. We need $x$ and $y$ to be in the range $[-100, 100]$:

$$-100 \leq 4p \leq 100 \implies -\frac{100}{4} \leq p \leq \frac{100}{4}$$

Since $p$ is an integer:

$$-25 \leq p \leq 25$$

So there are $25 - (-25) + 1 = 51$ possible values for $p$. We now count all the unique solutions. Note that the solutions $(3k, 4k)$ and $(-2m, 3m)$ and $(3p, 4p)$ and $(-2n, 3n)$ are the same sets of solutions, but counted separately. We must ensure we do not double-count any solutions. To avoid double-counting, we should count each unique $(x, y)$ pair only once. We have already counted each possible pair $(3k, 4k)$ and $(-2m, 3m)$ (or $(3p, 4p)$ and $(-2n, 3n)$) separately, but now we need to combine them. We note that $(0, 0)$ is a common solution for all cases. We also note that the sets $(3k, 4k)$ and $(-2m, 3m)$ do not overlap except for $(0, 0)$. So we count the total unique solutions as:

$$67 + 67 - 1 = 133$$

Thus, the total number of ordered pairs $(x, y)$ that satisfy the equation $12x^2 - xy - 6y^2 = 0$ with $x$ and $y$ in the range $[-100, 100]$ is $\boxed{133}$.

**Response from DisPPO:**

**1. Equation Analysis:**

The given equation is $12x^2 - xy - 6y^2 = 0$. This can be treated as a quadratic equation in terms of $x$. Rearrange it to:

$$12x^2 - xy - 6y^2 = 0$$

We can treat this as a quadratic in $x$, $12x^2 - xy - 6y^2 = 0$.

Using the quadratic formula $x = \frac{-b \pm \sqrt{b^2 - 4ac}}{2a}$, where $a = 12$, $b = -y$, and $c = -6y^2$, we get:

$$x = \frac{-(-y) \pm \sqrt{(-y)^2 - 4 \cdot 12 \cdot (-6y^2)}}{2 \cdot 12} = \frac{y \pm \sqrt{y^2 + 288y^2}}{24} = \frac{y \pm \sqrt{289y^2}}{24} = \frac{y \pm 17|y|}{24}$$

Since $\sqrt{289y^2} = 17|y|$, the solutions for $x$ are:

$$x = \frac{y + 17|y|}{24} \quad \text{or} \quad x = \frac{y - 17|y|}{24}$$

**2. Simplifying Solutions:**

We need to consider two cases based on the sign of $y$: - Case 1: If $y \geq 0$, then $|y| = y$. So the two solutions are:

$$x = \frac{y + 17y}{24} = \frac{18y}{24} = \frac{3y}{4} \quad \text{or} \quad x = \frac{y - 17y}{24} = \frac{-16y}{24} = -\frac{2y}{3}$$

- Case 2: If $y < 0$, then $|y| = -y$. So the two solutions are:

$$x = \frac{y + 17(-y)}{24} = \frac{y - 17y}{24} = \frac{-16y}{24} = -\frac{2y}{3} \quad \text{or} \quad x = \frac{y - 17(-y)}{24} = \frac{y + 17y}{24} = \frac{18y}{24} = \frac{3y}{4}$$

From both cases, the solutions for $x$ in terms of $y$ are $x = \frac{3y}{4}$ and $x = -\frac{2y}{3}$.

**3. Integer Constraints:**

Since $x$ must be an integer, for $x = \frac{3y}{4}$ and $x = -\frac{2y}{3}$, $y$ must be a multiple of 4 and 3 respectively.

- For $x = \frac{3y}{4}$, $y$ must be a multiple of 4. Let $y = 4k$, where $k$ is an integer. Then, $x = \frac{3(4k)}{4} = 3k$. Since $y = 4k$ and $-100 \le y \le 100$, we get $-100 \le 4k \le 100 \Rightarrow -25 \le k \le 25$. So $k$ ranges from -25 to 25 inclusive, which gives us 51 possible values for $k$. For each valid $k$, we get a corresponding $(x, y) = (3k, 4k)$.

- For $x = -\frac{2y}{3}$, $y$ must be a multiple of 3. Let $y = 3m$, where $m$ is an integer. Then, $x = -\frac{2(3m)}{3} = -2m$. Since $y = 3m$ and $-100 \le y \le 100$, we get $-100 \le 3m \le 100 \Rightarrow -\frac{100}{3} \le m \le \frac{100}{3}$. Since $m$ is an integer, $-33 \le m \le 33$. So $m$ ranges from -33 to 33 inclusive, which gives us 67 possible values for $m$. For each valid $m$, we get a corresponding $(x, y) = (-2m, 3m)$. We have to be careful not to double count any pairs. Note that the pair $(0, 0)$ (when $k = 0$ for the first case and $m = 0$ for the second case) is counted twice. So, we need to subtract 1 from the total.

**4. Counting the Pairs:** The total number of ordered pairs $(x, y)$ is the sum of the valid pairs for each case minus the double counted pair:

$$\text{Total pairs} = 51 + 67 - 1 = 117$$

Putting the final answer in the boxed format:

$$\boxed{117}$$

**Case Study Analysis.** We further conduct a qualitative case study on an AIME2025 algebra problem to examine the behavioral differences between the base model, the PPO baseline, and our DisPPO method. The equation

$$12x^2 - xy - 6y^2 = 0$$

admits a clean factorization into two linear components,

$$(3x + 2y)(4x - 3y) = 0,$$

implying that all solutions lie on one of two lines. Parameterizing these solutions yields $(x, y) = (-2k, 3k)$ with $k \in [-33, 33] \cap \mathbb{Z}$ (67 solutions) and $(x, y) = (3m, 4m)$ with $m \in [-25, 25] \cap \mathbb{Z}$ (51 solutions). The two sets intersect only at $(0, 0)$, giving the correct total of $67 + 51 - 1 = 117$ integer pairs.

The three models display distinct qualitative behaviors.

*Base model:* Although it successfully factorizes the expression, its subsequent reasoning-to-code execution is unstable, leading to incorrect conditional logic and an undercounted final answer.

*PPO:* The PPO-trained model exhibits fragmented and branching reasoning. It unnecessarily splits the solution into numerous subcases, repeatedly counts equivalent parametrizations, and ultimately overcounts despite attempting de-duplication, yielding a substantially inflated answer.

*DisPPO:* In contrast, DisPPO produces a coherent end-to-end trajectory with an explicit global plan: it first reduces the quadratic constraint to two linear factors (equivalently, two lines), then uses a single, non-redundant parametrization per line, and finally enforces the box constraints $[-100, 100]^2$ at the level of the free integer parameter. Importantly, it avoids the failure modes observed in the other two baselines: it does not rely on fragile code snippets for counting, and it does not introduce unnecessary sign-based branching that can lead to duplicated counting. As a result, the reasoning remains internally consistent (each step is directly justified by the previous one), and the final count follows from a clean inclusion–exclusion argument at the unique intersection point $(0, 0)$.

This example illustrates that distributional value estimation improves not only final accuracy but also the internal consistency of long-horizon reasoning trajectories.

