# OpenReview forum: "DisPPO: Quantile-Based Distributional Reinforcement Learning for Large Language Models"
_ICML.cc/2026/Conference — ICML 2026 regular_

### Official Review · Reviewer_jNBF · 2026-02-27

**Soundness:** 3
**Presentation:** 3
**Significance:** 3
**Originality:** 3
**Overall Recommendation:** 5
**Confidence:** 2

**Summary:**

Reinforcement Learning (RL) stands as a pivotal technique for enhancing the reasoning capabilities of Large Language Models (LLMs). While traditional approaches such as Proximal Policy Optimization (PPO) rely on scalar value functions to estimate cumulative returns, they inherently overlook higher-order statistical information. This limitation often results in inaccurate value estimation and suboptimal credit assignment. To mitigate these challenges, this paper proposes a novel framework based on Distributional RL, which learns a full distributional value function rather than a single scalar expectation. Empirical results demonstrate that this approach yields significant performance improvements over conventional baselines.

**Compliance With Llm Reviewing Policy:**

Affirmed.

**Final Justification:**

This paper proposes a novel framework based on Distributional RL, which learns a full distributional value function rather than a single scalar expectation. Empirical results demonstrate that this approach yields significant performance improvements over conventional baselines.
(1) The proposed Distributional PPO framework effectively incorporates higher-order statistical information into the value estimation process.
(2) The paper provides a rigorous theoretical analysis underpinning the proposed method. Notably, it establishes formal convergence guarantees, which lend strong mathematical credibility to the algorithm's stability and reliability during the optimization process.
(3) Comprehensive experiments spanning multiple domains—including mathematical reasoning, SQL code generation, and multi-turn tool-augmented reasoning—demonstrate the method's efficacy and robustness.
The rebuttal has addressed my questions, so I have set the Overall Recommendation score to Accept.

**Key Questions For Authors:**

Is N = 51 truly the optimal hyperparameter for the DisPPO framework, or was this value chosen empirically without rigorous validation?

**Limitations:**

yes

**Strengths And Weaknesses:**

### Strengths
(1) The proposed Distributional PPO framework effectively incorporates higher-order statistical information into the value estimation process.

(2) The paper provides a rigorous theoretical analysis underpinning the proposed method. Notably, it establishes formal convergence guarantees, which lends strong mathematical credibility to the algorithm's stability and reliability during the optimization process.

(3) Comprehensive experiments spanning multiple domains—including mathematical reasoning, SQL code generation, and multi-turn tool-augmented reasoning—demonstrate the method's efficacy and robustness.

### Weaknesses
(1) The implementation of Distributional PPO inherently incurs higher computational overhead due to the necessity of estimating a full distributional value function rather than a scalar expectation. The manuscript currently lacks a quantitative analysis of this trade-off. It is essential to explicitly report the additional computational costs (e.g., training time per step, memory footprint, and inference latency) relative to the standard PPO baseline. Without this data, the practical efficiency and scalability of the proposed method remain unclear.

(2)
In Line 373, the authors specify the use of N = 51 quantiles for the DisPPO framework. However, the paper does not provide an ablation study or sensitivity analysis regarding this critical hyperparameter. How does varying N impact reasoning performance and convergence stability? A detailed investigation into the relationship between the number of quantiles and model efficacy is necessary to justify the chosen configuration and to guide future practitioners in optimal hyperparameter selection.

---

> ### Author Rebuttal · Authors · 2026-03-31
>
> We thank the reviewer for the careful feedback on efficiency and hyperparameter sensitivity. Below we respond to the main concerns in turn.
>
> # Response to W1:
>
> We agree that a quantitative characterization of the computational trade-off is important. Our submission **already includes a FLOPs-based analysis**, and we will further clarify the wall-clock efficiency.
>
> In **Section 6.1** and **Appendix L (Tables 16–19)**, we report that DisPPO introduces only about 3.6%–4.0% additional training FLOPs over PPO, with no extra inference-time FLOPs, since only the actor is used at test time.
>
> Complementing this, under the same mathematical reasoning setup (Qwen2.5-Math-7B) and the same synchronous VERL framework on 4 nodes / 32 GPUs, the average runtime is 483.23 s for GRPO, 751.41 s for PPO, and 834.52 s for DisPPO. This corresponds to 1.11× PPO and 1.73× GRPO. Relative to PPO, the added cost is moderate (+11% runtime), while it is higher compared to critic-free GRPO. Importantly, although value-based methods introduce an additional critic model compared to value-free approaches, the training time does not double, since the critic does not require autoregressive generation.
>
> We clarify that the additional cost comes solely from replacing the scalar critic head with a quantile-based distributional critic, while leaving the actor and policy optimization pipeline unchanged.
>
> # Response to W2 (merged with Q1):
>
> We agree that the choice of N should be justified empirically. **Appendix H.2 (Table 12) already includes an ablation**on SQL generation with N in {17, 35, 51, 101}. The results improve consistently from 77.1 (N=17) to 77.9 (N=35) to 78.9 (N=51), while the gain beyond 51 is marginal (79.0 at N=101). Thus, within the tested range, N=51 offers the best performance-efficiency trade-off and was selected empirically rather than chosen arbitrarily.
>
> At the same time, we agree that the current evidence is task-specific and does not by itself establish a universally optimal N or provide a full convergence-stability study across all settings. We will revise the paper to make this scope explicit and to present the sensitivity study more prominently in the main text.

---

> > ### Author Rebuttal · Reviewer_jNBF · 2026-04-01
> >
> > Thank you f or the response. I'll maintain my positive score 5.

---

> > > ### Author Response · Authors · 2026-04-02
> > >
> > > Thank you for the clarification and for maintaining the positive score. I’m glad the concerns have been addressed.

---

### Official Review · Reviewer_UbZv · 2026-03-03

**Soundness:** 1
**Presentation:** 2
**Significance:** 2
**Originality:** 2
**Overall Recommendation:** 2
**Confidence:** 4

**Summary:**

The paper proposes a distributional PPO method to train LLMs. The authors use  quantile based distributional critic to replace the original scalar critic in PPO. Based on the previous literature, they also introduce GAE to the distributional critic and provide corresponding theoretical results. The authors run experiments on math reasoning, SQL code generation and tool calling. The proposed method shows the marginal improvement over baselines.

**Compliance With Llm Reviewing Policy:**

Affirmed.

**Key Questions For Authors:**

- How many rollouts per query do you generate in DisPPO?
- Can you compare the training time and memory usage of DisPPO and other critic-free methods?

**Limitations:**

yes

**Strengths And Weaknesses:**

Strengths:
- The idea of leveraging distributional RL (specifically, distributional critic) to capture the uncertainty in LLM reasoning is promising.

Weaknesses:
- The novelty of this paper is limited. The idea of QR critic, the surrogate function of Wasserstein distance (i.e., the huber loss), and the network design (distributional head) are all mainly derived from the previous works. While the authors introduces a distributional version of GAE, overall the novelty is very incremental.
- Most importantly, the distributional learning is not very meaningful in this paper's setting:
  - The main objective of distributional critic is to capture the uncertainty which can be from either environment stochasticity ($s,a\rightarrow$ different $s'$) or policy stochasticity. In token-based MDP in LLM, the env is actually deterministic. So here we only consider policy stochasticity.
  - However, the implementation of proposed method actually doesn't induce large policy stochasticity: in rollout, the LLM generates one or multiple answers for one query (initial state), but it won't generate multiple continuations for some middle states. Therefore, except for the initial state, most states have **only one** corresponding rollout and they have only one target in bellman update $Z(s) \leftarrow r + \gamma Z(s')$. In other words, there is neither environment stochasticity nor policy stochasticity in distributional learning. For most states, DisPPO is simply propagating the uncertainty of $Z(s')$ (which is only from the NN initialization) to $Z(s)$, which has no significant difference from standard PPO.
  - Actually, if you want to truly capture policy stochasticity (as QR-DQN does), you need to start from each state in the trajectory and rollout for multiple times, which is similar to off-policy learning but it requires much larger computation.
- Compared to current critic-free methods (GRPO and its variants), DisPPO requires much more computation and memory in practice. Meanwhile, as discussed above, the distributional learning mainly works for the initial states in DisPPO, but GRPO is also able to learn the distributional value of the initial state by multiple rollouts. So I don't see any significant advantage of DisPPO over critic-free methods.
- In experiment, while authors show that DisPPO exceeds the baselines, the advantage is relatively marginal.

---

> ### Author Rebuttal · Authors · 2026-03-31
>
> We thank the reviewer for the careful and important feedback.
> # Response to W2/W3:
> We appreciate the reviewer’s observation regarding the deterministic nature of LLM transitions. We would like to clarify how DisPPO captures uncertainty induced by stochastic policy in this specific setting.
> 1. Defining Uncertainty
> The reviewer is correct that transitions $P(s'|s, a)$ in LLMs are deterministic. Consequently, we focus exclusively on policy-induced stochasticity.
>  - The objective of the distributional critic $Z^\pi(s)$ is to model the distribution of the return $G = \sum_{t} \gamma^t r_t$. Even in a deterministic environment, $G$ is a random variable induced by the stochastic policy $\pi$.
>  - Bimodal Intuition: In the RLVR, where rewards are binary ($r \in \{0, 1\}$) and there is no discounting ($\gamma = 1$), the return $G$ for any trajectory is inherently discrete ($0$ or $1$). Thus, for any intermediate state, the target distribution is a bimodal (Bernoulli) distribution. Capturing this binary "uncertainty of success" is far more representative of the reasoning process than a collapsed scalar mean.
>
> 2. Capturing Distributions without Middle-State Continuations: Generalization vs. Multiple Rollouts
>
>    The reviewer argues that because each prefix has only one rollout, there is "no stochasticity to capture." We believe this reflects a **lookup table** view of RL that overlooks the power of **function approximation**:
>
>  - In Deep RL, the critic is a neural network $Z_\theta(s)$ that learns from state features, not unique state IDs. While a specific prefix might appear once in a batch, the network observes thousands of **semantically and structurally similar prefixes** across the training set. By aggregating these shared features, the neural network effectively integrates information from multiple trajectories. This allows it to learn the full return distribution for a "type" of state, even if each individual state is visited once.
> - Crucially, **if the reviewer’s logic—that one rollout per state prevents learning—were true, then standard PPO would also fail to estimate the mean value $V(s)$ in LLMs**. However the success of PPO proves that TD-learning through function approximation successfully captures policy-induced variance.
> - Empirical Verification: Capturing Unseen Distributions
>   The results in Table 5 provide direct empirical evidence that DisPPO captures the underlying return distribution even without multi-path sampling during training. In Table 5, the ground-truth MC returns are calculated by performing 256 continuations from each **intermediate** prefix. During training, DisPPO **never** observes multiple continuations from these middle states. However, Table 5 shows that DisPPO achieves a consistently lower MAE than standard PPO across all relative positions.
>
> # Response to W1:
> We appreciate the feedback but respectfully disagree that the contribution is merely incremental.
> 1. Technical Novelty: Convergent Distributional $\lambda$-Return
>    While previous works (e.g., QR-DQN) are limited to TD(0), we derived a distributional version of the $\lambda$-return operator ($\mathcal{T}^{\pi, \lambda}$).This extension is essential for the stability of PPO in large-scale LLM training, where variance reduction via GAE is critical.  We provide first-of-their-kind proofs for the contraction mapping properties of this operator (**Theorems 4.1 & 4.2**), ensuring the mathematical soundness and convergence of our framework—a theoretical foundation missing in "incremental" architectural changes.
> 2. Domain Novelty: Distributional RL for LLM Reasoning
> - Our work is the first systematic effort to apply distributional RL to the deterministic-transition, bimodal-reward landscape of LLM reasoning (RLVR).
> - Beyond Stochastic Environments: We challenge the conventional wisdom that distributional RL is only for heavy-tailed environment noise. We demonstrate that even in deterministic LLM environments, the policy-induced bimodal uncertainty ($r \in \{0, 1\}, \gamma=1$) makes distributional modeling superior to scalar regression.
>
> 3. Empirical Result:
>
>    We establish a key empirical result that prior works have not: distributional supervision leads to more accurate mean value estimation than direct scalar regression in Table 5.
>
> # Response to W4:
> Prior work (VC-PPO, VAPO) shows that value-based methods benefit from token-level feedback, improved credit assignment, and better sample efficiency, leading to stronger performance. We view DisPPO as a moderate-overhead improvement over PPO, not a replacement for value-free methods.
> # Response to Q1:
> DisPPO uses the same rollout budget as PPO (16 for reasoning, 8 for SQL; Appendix C, Tables 6–8).
> # Response to Q2&W5:
> Q2: Please see our detailed response to Reviewer jNBF in Note g0PLTinuHD. (Response to W1)
>
> W5: Please see our detailed response to Reviewer 5Gna in Note HTeNRPktQ7. (Response to W2&Q2)
>
>  [1] Yue et al., VAPO, 2025. [2] Yuan et al., VC-PPO, 2025.

---

> > ### Author Rebuttal · Reviewer_UbZv · 2026-04-04
> >
> > Thank you for the detailed clarification. I agree that function approximation can generalize across similar prefixes. Here are some comments:
> > - My main concern remains about how **effectively** this distribution is learned in the proposed setting. In PPO-style LLM training, most intermediate states are associated with only a single sampled continuation, which provides very limited direct supervision for estimating a full return distribution. While generalization from function approximation may help, this makes the learned distribution largely dependent on inductive bias rather than strongly grounded multi-sample evidence from the same state.
> > - In this sense, DisPPO appears closer to standard PPO with a different critic parameterization (i.e., a quantile head) rather than a setting where distributional RL can fully exploit its typical advantages. Importantly, single-sample Monte Carlo works well for scalar PPO because it only needs to regress the **mean**. **However, the information requirement is much higher for a full distribution.** A single sampled continuation from a state does not directly identify a nontrivial conditional return distribution at that state.
> > - Additionally, in the RLVR setting with binary terminal rewards, the return distribution is essentially Bernoulli and therefore fully characterized by its mean. As a result, a scalar value function in PPO is already sufficient to represent the underlying uncertainty, which makes it even more unclear what fundamental advantage the distributional learning provides beyond a quantile critic head.

---

> > > ### Author Response · Authors · 2026-04-04
> > >
> > > Dear Reviewer UbZv,
> > >
> > > We sincerely thank you for these insightful comments. Your theoretical observation—that a Bernoulli distribution is mathematically characterized by its mean—is a deep point. In fact, this aligns precisely with the initial concern raised by **Reviewer Au22**, who also questioned why a distributional critic should outperform a scalar critic and sought clarification on the superior MAE performance of DisPPO in Table 5. Our subsequent clarification and detailed analysis successfully resolved these doubts.
> > >
> > > The reason is that the distributional critic and scalar critic lead to  different optimization behaviors in neural networks.  A scalar critic (MSE) attempts to regress the global mean, which often results in a non-informative value (e.g., $0.5$ for a high-uncertainty path). In contrast, DisPPO models the full distribution, forcing the network to preserve the uncertainty structure—specifically the "gap" between success and failure.
> > >
> > > To provide a more granular understanding of the value estimation performance, we performed a post-hoc analysis of the data underlying Table 5 by re-binning the Monte Carlo (MC) ground-truth targets into three regions in our response to Reviewer Au22. The resulting pattern is clearer: DisPPO performs better at both ends, while PPO is better only in a relatively small middle region.
> > >
> > > | MC target bin | Share | DisPPO MAE | PPO MAE | Diff (D-P) |
> > > | :------------ | :---- | :--------- | :------ | :--------- |
> > > | [0.0, 0.3)    | 74.5% | 0.3242     | 0.3446  | -0.020     |
> > > | [0.3, 0.7)    | 5.9%  | 0.2346     | 0.1723  | +0.062     |
> > > | [0.7, 1.0]    | 19.6% | 0.1573     | 0.1714  | -0.014     |
> > >
> > >  This is exactly the pattern we would expect if the main benefit of DisPPO is to preserve boundary sensitivity rather than regress toward the middle. For reasoning tasks with binary outcomes, precision at the boundaries is far more consequential than precision near the midpoint. Reasoning often exhibits a cliff effect, where a single token can flip the entire outcome from success to failure. A critic that is more accurate in these boundary regions therefore provides higher-fidelity advantage signals for policy updates. Our results suggest that this boundary sensitivity is the primary driver of DisPPO's improved reasoning accuracy on complex benchmarks such as AIME25. Please see our reply to  reviewer Au22 for further discussion.
> > >
> > >
> > > Generalization to Complex Reward Landscapes: The advantages of DisPPO become even more pronounced—as we expected—as the reward landscape increases in complexity. In our response to **Reviewer 5Gna**, we demonstrated that DisPPO excels in standard RLHF settings with continuous, noisy reward signals (using the UltraFeedback dataset and a learned reward model as the stepup in [1] ):.
> > >
> > > ## Table 2: Standard RLHF comparison table
> > >
> > > | Methods | AlpacaEval 2 (WR) | Arena-Hard | MT-Bench |
> > > | :------ | :---------------- | :--------- | :------- |
> > > | DisPPO  | **30.14**         | **19.3**   | **7.41** |
> > > | PPO     | 20.77             | 18.3       | 7.18     |
> > > | VAPO    | 23.78             | 18.5       | 7.23     |
> > >
> > >
> > >
> > > [1] Yin et al., *Segmenting Text and Learning Their Rewards for Improved RLHF in Language Model*, 2025.

---

### Official Review · Reviewer_Au22 · 2026-03-11

**Soundness:** 3
**Presentation:** 3
**Significance:** 2
**Originality:** 2
**Overall Recommendation:** 5
**Confidence:** 5

**Summary:**

This paper introduces DisPPO as an algorithm for reinforcement learning for LLMs, a modification of the PPO algorithm by using a quantile critic architecture. In particular, their critic approximates the distribution of possible returns, and they then use the mean of this distribution when training the actor. They train the critic using a distributional GAE loss, which they justify by demonstrating that the $\lambda$-return version of the projected distributional Bellman operator is a convergence in the maximal $W_\infty$ metric. They compare their method against PPO and DAPO across a range of LLM RL benchmarks, in particular mathematical reasoning, SQL generation, and multi-turn tool use.

**Compliance With Llm Reviewing Policy:**

Affirmed.

**Final Justification:**

My final recommendation is an accept -- in their rebuttal, the authors addressed all of my questions, and I am happy with their additional experiment. I have read the other reviews and discussions, and stand by my decision.

**Key Questions For Authors:**

- Can you provide any insight onto what is causing the improved MAE of quantile mean estimation in this setting?
- Why is Theorem 4.2 in terms of $W_\infty$ while Theorem 4.1 is in terms of $W_p$? Does Theorem 4.2 not hold for $p \ne \infty$, or does the proof simply not follow as straightforwardly?
- In Table 1, it states that the advantage in DisPPO is a vector. Should this be modified to say a scalar? From my understanding of Section 3.3 the algorithm takes the mean of quantiles before passing it to the actor, so the advantages should be unchanged.

**Limitations:**

yes

**Strengths And Weaknesses:**

**Strengths**
- The paper is well-written, and explains their method well. I enjoyed the discussion of different architecture choices for the quantile parameterization.
- Although the proof technique itself is not entirely novel, it is nice to include the theoretical results on the distributional GAE contraction as theoretical support for the method.
- The algorithm provides reasonably strong improvements over PPO/DAPO, and I believe that this can lead to future papers working on distributional RL approaches to LLM RL.

**Weaknesses**
- There is limited novelty to the actual method (indeed, it is essentially applying a well-known distributional RL method to LLM RL). That said, I do think there is value in demonstrating that distributional RL can improve LLM RL, and the paper provides useful empirical evidence in that direction. My main concern is that there is relatively little investigation into why the distributional methods outperform here. I appreciate that Table 5 demonstrates that the performance gains can be attributed to improved MAE of value estimation, but I am principally interested in *why* this outperformance may be happening.
  - For more detail, this question has been studied in classical RL as well. The most relevant work is "The statistical benefits of quantile temporal-difference learning for value estimation" by Rowland et al., 2023 (and to a lesser extent, Kastner et al., 2025 "Categorical distributional reinforcement learning with Kullback-Leibler divergence: convergence and asymptotics" which analyzed this question in the case of categorical distributions).
  - Both of these papers came to a similar conclusion: the outperformance of distributional methods over pure mean estimation is primarily when the return distributions are heavy-tailed (e.g. heavier than Gaussian).
  - With this in mind, the results of Table 5 (that quantile-based estimation outperforms MSE) surprised me in this setting, as if I understand correctly all returns are in $[0,1]$. I think that if this point can be further investigated, and if you can share some insights/experimental results understanding why distributional methods help in LLM-based RL, the contribution of this work will be improved.
- A lot of related work in the theory of distributional RL is missing. For example, the appendix states "[...] the theoretical foundations of distributional RL are canonically established on state-action value distributions Z(s, a)." which is arguably untrue, as most of the theory of distributional RL has been studied in the policy evaluation setting. I think that the following papers should be cited at the very least: Bellemare et al, 2023 "Distributional Reinforcement Learning" (textbook);  Rowland et al., 2023 "An analysis of quantile temporal-difference learning", Rowland et al., 2023 "The statistical benefits of quantile temporal-difference learning for value estimation".  Similarly, the proofs can probably be tightened if they are modified to operate over distributions rather than random variables.

**Minor**
- Q♯ (Zhou et al., 2025) should probably be discussed as related work.
- “value-freed-based-baselines” -> “value-free baselines”
- In Equation 33, $\mathcal{L}_{pop}$ should depend on $\tau$.

---

> ### Author Rebuttal · Authors · 2026-03-31
>
> We thank the reviewer for the careful and technically detailed feedback. Below we respond to the main concerns and questions in turn.
>
> # Response to W1 and Q1:
>
> We thank the reviewer for the insightful comment and for pointing us to the foundational work by Rowland et al. (2023) and Kastner et al. (2025). We agree that in classical RL, the advantages of Quantile Temporal-Difference (QTD) learning are often attributed to heavy-tailed return distributions. However, we argue that LLM-based reasoning (RLVR) presents a unique bimodal distribution that provides a different but equally compelling case for distributional RL. The following are some of our insights.
>
> In the RLVR setting, two factors fundamentally shape the value distribution:
>
> 1. The final reward is strictly $r \in \{0, 1\}$ (success or failure).
> 2. A common and illustrative setting is $\gamma = 1$ and $\lambda = 1$ in value learning.
>
> Consequently, the ground-truth return $G$ for any trajectory is discrete and binary. At any intermediate reasoning step, the ideal value function should therefore represent a Bernoulli distribution. A scalar critic (MSE) attempts to regress the mean ($p$), which is often a non-informative value (e.g., $0.5$ for a half-correct path). In contrast, DisPPO models the full distribution, forcing the network to maintain the uncertainty structure, i.e., the gap between success and failure.
>
> To validate this, we re-binned the Monte Carlo (MC) ground-truth targets into coarse low/middle/high regions. The resulting pattern is clearer: DisPPO performs better at both ends, while PPO is better only in a relatively small middle region.
>
> | MC target bin | Share | DisPPO MAE | PPO MAE | Diff (D-P) |
> | :- | -: | -: | -: | -: |
> | [0.0, 0.3)|74.5% | 0.3242 |  0.3446 |-0.020 |
> | [0.3, 0.7)|5.9% | 0.2346 |  0.1723 |+0.062 |
> | [0.7, 1.0]|19.6% |0.1573 |  0.1714 |-0.014 |
>
> Aggregating these bins, DisPPO wins in the two boundary regions, $[0.0, 0.3)$ and $[0.7, 1.0]$, whereas PPO wins only in the middle region $[0.3, 0.7)$. This is exactly the pattern we would expect if the main benefit of DisPPO is to preserve boundary sensitivity rather than regress toward the middle.
>
> The reviewer may notice that the sample distribution is skewed toward the boundaries. We emphasize that this is not a statistical artifact, but a natural reflection of LLM reasoning dynamics: as training progresses, trajectories increasingly become either clearly correct or clearly flawed. Scalar PPO (MSE) performs best when the target is near the expectation of a mixed-success distribution, because MSE is optimized to recover the center of mass. However, for reasoning tasks with binary outcomes, precision at the boundaries is far more consequential than precision near the midpoint. Reasoning often exhibits a cliff effect, where a single token can flip the entire outcome from success to failure. A critic that is more accurate in these boundary regions therefore provides higher-fidelity advantage signals for policy updates. Our results suggest that this boundary sensitivity is the primary driver of DisPPO's improved reasoning accuracy on complex benchmarks such as AIME25.
>
> # Response to Q2:
>
> Regarding the reviewer's question on whether Theorem 4.2 holds for $p < \infty$: The answer is generally no.  Lemma 5 of QR-DQN (arxiv version) [1] provides an explicit counter-example demonstrating that $\Pi_{W_1}\mathcal{T}^{\pi}$ can be expansive in the metric $\bar{d}_p$ when $p < \infty$, meaning it may actually increase the distance between two distributions after the projection step. This counter-example remains valid in our setting, as our distributional $\lambda$-Bellman operator $\mathcal{T}^{\pi, \lambda}$ recovers the standard distributional Bellman operator when $\lambda = 0$. Due to space constraints, we kindly refer the reviewer to Lemma 5 in the Appendix of the arXiv version of QR-DQN.
>
> # Response to Q3:
>
> We thank the reviewer for the careful reading and for catching this typo. You are absolutely correct—while the critic in DisPPO learns a distributional representation, the advantage used for the policy update is indeed a scalar. We will correct Table 1 in the revised manuscript.
>
> # Response to W2:
>
> We agree that the theory-side related-work coverage is incomplete and that the wording in Appendix A is too strong. We will revise this wording, add the missing references suggested by the reviewer (including Bellemare et al., 2023 and the Rowland et al. papers), better situate our presentation within the broader policy-evaluation literature in distributional RL, and clean up the proof presentation.
>
> # Response to Minor:
>
> We will add Q♯ (Zhou et al., 2025), fix the typo "value-freed-based-baselines", and make the dependence of $\(\mathcal{L}_{pop}\)
> $ on $\(\tau\)$ explicit in Eq. 33.
>
> [1] Dabney et al., Distributional Reinforcement Learning with Quantile Regression, AAAI 2018.

---

> > ### Author Rebuttal · Reviewer_Au22 · 2026-04-04
> >
> > I thank the authors for their work on the rebuttal, all of my questions and concerns have been addressed. I am particularly happy with their experiment on binning the targets and analyzing the MAE of the methods in each bin, I believe this improves my intuition and would be illustrative for readers of the paper in general. I am happy to increase my score to a 5.

---

> > > ### Author Response · Authors · 2026-04-04
> > >
> > > Dear Reviewer Au22，
> > >
> > > We sincerely thank you for your positive feedback and for the decision to increase the score. We are very glad that the more granular analysis derived from Table 5 (binning the targets) helped clarify the intuition behind our methods.
> > >
> > > Just a gentle reminder: as the system still shows the previous score 4, we would appreciate it if you could update the score entry to a 5 as you mentioned. Many thanks!

---

### Official Review · Reviewer_5Gna · 2026-03-13

**Soundness:** 2
**Presentation:** 3
**Significance:** 3
**Originality:** 3
**Overall Recommendation:** 4
**Confidence:** 3

**Summary:**

This paper enhances the reasoning capabilities of LLM through distributinoal reinforcement learning. It proposes the DisPPO algorithm, aiming to improve upon the existing PPO framework. While PPO only estimates the mean of cumulative returns, DisPPO leverages higher-order statistical information and addresses this limitation by modeling the full return distribution. The Ctiric in DisPPO predicts a complete return distribution, and to extend GAE into the distributional RL setting, the authors introduce D-GAE. The authors evaluated the method across multiple benchmarks using Llama and Qwen models, demonstrating consistent performance improvements.

**Compliance With Llm Reviewing Policy:**

Affirmed.

**Final Justification:**

Although the authors' rebuttal has addressed my concerns, my overall evaluation of the paper remains unchanged. Therefore, I am maintaining my score 4.

**Key Questions For Authors:**

1. How to justify the relative superiority of DisPPO's distributional approach when the paper excludes comparisons with methods specifically designed to mitigate PPO value estimation bias, such as VAPO? Without these benchmarks, it is unclear if the proposed method offers advantages beyond existing value-regularization frameworks.
2. Do the incremental performance gains, such as the 1.9% average increase in mathematical reasoning (from 43.1% to 45.0%), significantly exceed optimization noise?
3. How does DisPPO generalize to standard RLHF scenarios involving continuous and potentially noisy human preference rewards? Since the current evaluation is restricted to reasoning tasks with verifiable binary or sparse outcomes, there is a lack of evidence regarding the method's efficacy in broader, less-structured reward landscapes.

**Limitations:**

While the authors provide solid theoretical guarantees and thorough test results for DisPPO on fixed-answer reasoning tasks, they do not systematically lay out the method’s technical limitations. In addition, the paper’s discussion of ethical and social impacts is too brief.

**Strengths And Weaknesses:**

**Strengths**
- It addresses the point of scalar value estimation bias in LLM PPO training, with a highly practical method that is fully compatible with existing PPO pipelines, has negligible extra overhead, and offers a generalizable new paradigm for enhancing LLM RL training.
- The work is technically rigorous, with complete theoretical proof of the update operator’s contraction property to guarantee convergence, paired with comprehensive, fair experiments across multiple reasoning tasks, model families, and targeted ablation studies to fully validate the method’s effectiveness and stability
- This well-written, highly readable paper follows a clear logical narrative from problem motivation to theory and empirical validation, with complete pseudocode, detailed hyperparameters, and intuitive visualizations.

**Weakness**

- The paper lacks comparative experiments with recent SOTA methods targeting PPO value estimation bias, leaving the method’s generalizability and relative superiority incompletely validated.
- Marginal Performance Gains The empirical improvements across benchmarks are relatively modest. For mathematical reasoning, the average `avg@k` increase is only 1.9% over the standard PPO baseline. It is unclear whether these incremental gains justify the structural overhead or are simply within the range of optimization noise.

---

> ### Author Rebuttal · Authors · 2026-03-31
>
> We thank the reviewer for the careful and constructive feedback. Below we respond to the main concerns and questions in turn.
> # Response to W1 & Q1:
> To address this concern more directly, we additionally ran VAPO during the rebuttal period. DisPPO consistently outperforms VAPO across the evaluated benchmarks; detailed results are provided in the Table 1 below. These results further support that the gains of DisPPO do not come from comparing against weak PPO baselines, but from improving value learning within an already strong PPO training pipeline.
>
> We would also like to clarify that our original submission already follows a strong PPO training recipe consistent with recent large-scale RL-for-reasoning systems. **As stated in Section 6.1 (line 320)**, our PPO baselines are already strong, **incorporating key optimization techniques from VAPO**, including Clip-Higher, Token-level Loss, Critic Pretraining, and Group Sampling. The entropy-regularized PPO variant is applied on top of this strengthened PPO baseline, rather than vanilla PPO, and we also include entropy-regularized DAPO as a strong baseline.
>
> ## Table 1：Avg@k results (mean ± std) over three random seeds
>
> | Method | AIME24 |College |MATH500|Minerva| OlympiadBench|AIME25|    Avg.    |
> | ------ | :--------: | :--------: | :--------: | :--------: | :-----------: | :--------: | :--------: |
> | DisPPO | 36.4 ± 1.4 | 42.8 ± 0.3 | 82.2 ± 0.1 | 46.7 ± 0.2 |  44.1 ± 0.3  | 19.3 ± 0.5 | 45.3 ± 0.3 |
> | VAPO   | 33.0 ± 1.9 | 41.9 ± 0.3 | 81.3 ± 0.2 | 45.0 ± 0.2 |  42.7 ± 0.4   | 18.0 ± 0.4 | 43.6 ± 0.3 |
> | PPO    | 32.5 ± 1.1 | 42.0 ± 0.4 | 81.1 ± 0.1 | 44.5 ± 0.2 |  43.0 ± 0.3   | 16.6 ± 0.4 | 43.3 ± 0.2 |
>
> Note: mean ± std over three repeated runs with different random seeds.
>
> # Response to Q3:
>
> To address this concern, we additionally conduct experiments during the rebuttal period in a standard reward-model-based RLHF setup using LLaMA3-8B. Specifically, we used the open-source PPO RLHF setup of Yin et al. [1] on UltraFeedback preference data, where a learned reward model assigns a scalar reward only at EOS. This setting is substantially less structured than our original benchmarks: rewards are produced by a learned reward model trained from human-preference data, and are therefore continuous/noisy rather than verifiable binary outcomes.
>
> For a controlled comparison, we keep the backbone model, reward model, dataset, and PPO training recipe unchanged, and replace only the scalar critic in PPO with a distributional critic. Under this setting, DisPPO improves AlpacaEval 2 win rate from 20.77 to 30.14, Arena-Hard from 18.3 to 19.3, and MT-Bench from 7.18 to 7.41 over the PPO baseline.
>
>  ## Table 2: Standard RLHF comparison table
>
> | Methods| AlpacaEval 2 (WR) | Arena-Hard | MT-Bench |
> | - | :-: | :-: | :-: |
> | DisPPO  |     **30.14**     |  **19.3**  | **7.41** |
> | PPO     |       20.77       |    18.3    |   7.18   |
> | VAPO     |       23.78       |    18.5    |   7.23   |
>
> These additional results suggest that the benefit of DisPPO is not restricted to verifiable reasoning tasks, and that distributional value modeling can also be useful in standard reward-model-based RLHF settings with learned, continuous, and noisy reward signals.
>
> # Response to W2&Q2
>
> To address this concern, we additionally repeat the mathematical reasoning experiments for DisPPO, VAPO, and PPO with three independent random seeds.
>
> As shown in Table 1 above in this rebuttal, DisPPO achieves the best average score across all six benchmarks (45.3), compared with 43.6 for VAPO and 43.3 for PPO. The margins (+1.7 over VAPO and +2.0 over PPO) are significantly larger than the reported standard deviations, suggesting that the gain **is stable rather than artifacts of favorable random seeds**.
>
> This conclusion is also consistent with the rest of our evidence: on mathematical reasoning, DisPPO improves both avg@k (+1.9) and pass@k (+3.9), and the improvement is also positive on SQL generation (+2.6 average) and multi-turn tool-augmented reasoning (+1.1 average). Moreover, the performance gain is accompanied by lower value prediction error (Table 5), suggesting improved critic learning rather than random variation alone.
>
> We will further extend this analysis in the revision by including repeated-run results for additional benchmarks to strengthen the stability evaluation.
>
>
> Reference: [1] Yin et al., *Segmenting Text and Learning Their Rewards for Improved RLHF in Language Model*, 2025.

---

> > ### Author Rebuttal · Reviewer_5Gna · 2026-04-04
> >
> > I thank the authors for the additional experiments and replies. Please ensure that the experiments provided in the rebuttal are incorporated into the paper, as this will enhance its completeness—especially Table 2: Standard RLHF Comparison table. My overall evaluation of the paper remains unchanged, and I will keep my current score.

---

> > > ### Author Response · Authors · 2026-04-04
> > >
> > > Thank you for your positive feedback and for recognizing the value of the additional experiments provided during the rebuttal.

---

### Decision · Program_Chairs · 2026-04-30

**Decision:**

Accept (regular)

**Comment:**

The paper addresses a timely and relevant question: whether distributional RL can improve PPO-style post-training for LLMs by replacing the standard scalar critic with a quantile-based distributional critic. Reviewers generally agreed that the paper is clearly written, technically sound, and supported by meaningful theoretical analysis. Several reviewers also found the empirical study reasonably broad, and the rebuttal addressed many of the initial concerns.

Reviewer UbZv raised the most substantive remaining concern, namely whether the observed gains reflect genuinely meaningful distributional value learning under the single-rollout PPO regime, or primarily a different critic parameterization. I consider this a valid limitation of the paper, but not one that overturns the overall case for acceptance given the paper's technical soundness, theoretical support.

The paper does have limitations, including modest empirical gains, incremental novelty relative to prior distributional RL work, and a need for clearer reporting of efficiency tradeoffs. I highly encourage the author to further enhance the empirical analysis of the proposed method and significantly consider the review comments and revise their future version of the paper.

I recommend weak acceptance.